# Intercomparisons of Tracker v1.1 and four other ocean particle tracking software packages in the Regional Ocean Modeling System

Jilian Xiong[1], Parker MacCready[1]

[1]School of Oceanography, University of Washington, Seattle, WA 98195, USA

*Correspondence to*: Jilian Xiong (jxiong7@uw.edu; xiongjilian@gmail.com)

**Abstract.** Particle tracking is widely utilized to study transport features in a range of physical, chemical, and biological processes in oceanography. In this study, a new offline particle tracking package, Tracker v1.1, is introduced and its performance is evaluated in comparison to an online Eulerian dye, one online and three offline particle tracking software packages in a small high-resolution model domain and a large coarser model domain. It was found that both particle and dye approaches give similar results across

different model resolutions and domains when they were tracking the same water mass, as indicated by similar mean advection pathways and spatial distributions of dye and particles. The flexibility of offline particle tracking and its similarity against online dye and online particle tracking make it a useful tool to complement existing ocean circulation models. The new Tracker was shown to be a reliable particle tracking package to complement the Regional Ocean Modeling System (ROMS) with the advantages of platform independence and speed improvements especially in large model domains achieved by the nearest neighbor search

algorithm. Lastly, tradeoffs of computational efficiency, modifiability, and ease of use that can influence the choice of which package to use are explored. The main value of the present study is that the different particle and dye tracking codes were all run on the same model output, or within the model that generated the output. This allows some measure of intercomparison of the different tracking schemes, and we conclude that all choices that make each tracking package unique do not necessarily lead to very different results.

## 1 Introduction

Lagrangian particle tracking is a very common and useful tool, especially in the post-processing of existing oceanographic model runs (van Sebille et al., 2018), and is of great value in applied oceanography like pollutant dispersion (e.g., Havens et al., 2009; Nepstad et al., 2022), oil spills (e.g., Nordam et al., 2019), harmful algal blooms (e.g., Giddings et al., 2014; Rowe et al., 2016; Xiong et al., 2023), planktonic larvae (e.g., Brasseale et al., 2019; Garwood et al., 2022), marine plastics (e.g., Onink et al., 2021),

and search-and-rescue (e.g., Chen et al., 2012), to name a few. Particle trajectories can be computed "online" along with the velocity fields at every time step as a part of ocean circulation models, for instance, the built-in particle tracking module "floats" in the Regional Ocean Modeling System (ROMS, Shchepetkin and McWilliams, 2005; Melsom et al., 2022; Aijaz et al., 2024). The trajectories can also be computed "offline" using stored hydrodynamic model output (Dagestad et al., 2018). Generally, offline tracking is more frequently applied in the literature than online tracking given its flexibility, for example, in working with different

precalculated velocity fields, testing particle seeding strategies and particle behaviors (Dagestad et al., 2018; Nordam and Duran, 2020; Hunter et al., 2022).

Many offline particle tracking software packages have been developed for multiple applications in oceanography, e.g., OceanParcels (Lange and van Sebille, 2017), Ichthyop (Lett et al., 2008), TRACMASS (Döös et al., 2013), PaTATO (Fredj et al., 2016), TrackMPD (Jalon-Rojas et al., 2019), OceanTracker (Vennell et al., 2021), Deft3D-PART (Deltares, 2022), Ariane (Blanke

and Raynaud, 1997), and CMS (Paris et al., 2013). Several previous studies have compared one Lagrangian particle tracking model

with passive Eulerian dye experiments to evaluate how well the particle trajectories integrated in a Lagrangian framework can represent the dye spreading in an Eulerian framework (e.g., North et al., 2006; Wagner et al., 2019; Melsom et al., 2022; Nepstad et al., 2022). Yet few (e.g., Daher et al., 2020) have compared different particle tracking models since the tracking codes are often developed to work with separate ocean models or forcing file formats. It is challenging to draw conclusions by comparing the output from each of them. Given the increasing popularity of particle tracking techniques in studying ocean transport features, it is useful to evaluate the performance (e.g., its similarity to Eulerian dye transport and computation speed) of the popular particle tracking software packages that can be assessed in a uniform testbed, e.g., using the same ocean circulation model. Here, we utilized a realistic, circulation model LiveOcean (MacCready et al., 2021) to evaluate several publicly available and commonly used particle tracking software packages.

LiveOcean is built using ROMS and is a realistic numerical model of ocean circulation and biogeochemistry for the coastal and estuarine waters of the northern California Current System (MacCready et al., 2021). The model is run quasi-operationally, making three-day forecasts of currents and other water properties every day. It is widely used by a variety of stakeholders concerned with the effects of ocean acidification, hypoxia, harmful algal blooms, and larval transport on fisheries. The model configuration of LiveOcean evolved from many years of research and modelling work in the coastal waters of Oregon, Washington, and most of Vancouver Island and in the Salish Sea (Sutherland et al., 2011; Davis et al., 2014; Giddings et al., 2014; Siedlecki et al., 2015). More details of model setup and validation are given in the Supplement of MacCready et al. (2021). LiveOcean has an offline particle tracking code written in Python named "Tracker" (v1.1), which has been used to identify the source of estuarine inflow from continental shelves (Brasseale and MacCready, 2021) and track trajectories of the harmful species *Pseudo-nitzschia* in daily post-processing to assist resource managers to decide to open or close WA beaches for razor clam harvest in combination with beach sampling (Stone et al., 2022). A snapshot of particle trajectories in the daily forecast of LiveOcean on January 12, 2024, can be found in the supplement of the present study.

To further evaluate the performance of Tracker and conduct multiple particle tracking model evaluations, three offline tracking codes: LTRANS (Schlag and North, 2012), OpenDrift (Dagestad et al., 2018), and Particulator (Banas et al., 2009) were selected among other particle codes. We selected these three packages because they all can operate on the original velocity fields solved on an Arakawa "C" grid (Arakawa and Lamb, 1977) used by ROMS, facilitating the direct intercomparison without the need for re-gridding velocity. They span a representative range of common programming languages, Fortran, Python, and MATLAB, as well as a range of algorithm choices (Table 1). Besides intercomparisons among these offline particle tracking codes, online passive dye experiments are used as a benchmark to evaluate their performance. ROMS online particle tracking "floats" is also tested to supplement the comparisons. To facilitate the implementation of online dye and particle tracking, a new, nested hydrodynamic model that only covers the domain of Hood Canal (Figure 1b) was established using ROMS. The Hood Canal model has a uniform horizontal resolution of 200 m and shares the same 30 vertical layers with LiveOcean. The northern open boundary is interpolated from the LiveOcean large domain while all other forcings (river and atmospheric forcings) come from the same sources as LiveOcean. Freshwater discharge from an additional eight tiny rivers (Figure 1b) was added to improve the simulated salinity field in Hood Canal.

In this short paper, we made a series of tests of four offline and one online particle tracking software packages to evaluate to what extent they all produce the same answer and to what extent they can reproduce results consistent with a passive dye. The main purpose is to conduct the intercomparisons of some commonly used particle tracking codes in the same numerical simulations to explore the net effect of the many slightly different choices made by the different developers. The other four offline tracking codes

have been rigorously tested by their developers, and we present our own tests of vertical mixing for Tracker. When choosing a particle tracking code to use, modelers have many considerations. Will the code be easy to use with their model output? Will they be able to modify the code for their specific needs, e.g., introducing vertical particle behavior? Will it run fast enough? Finally, a modeler should have some confidence that regardless of which code they choose the results will be reasonably similar for all the choices. The goal of this intercomparison is primarily to address this final issue of confidence. We also kept track of the computational efficiency and discussed ease of use of all tracking codes to provide practical guidance about tradeoffs for other researchers.

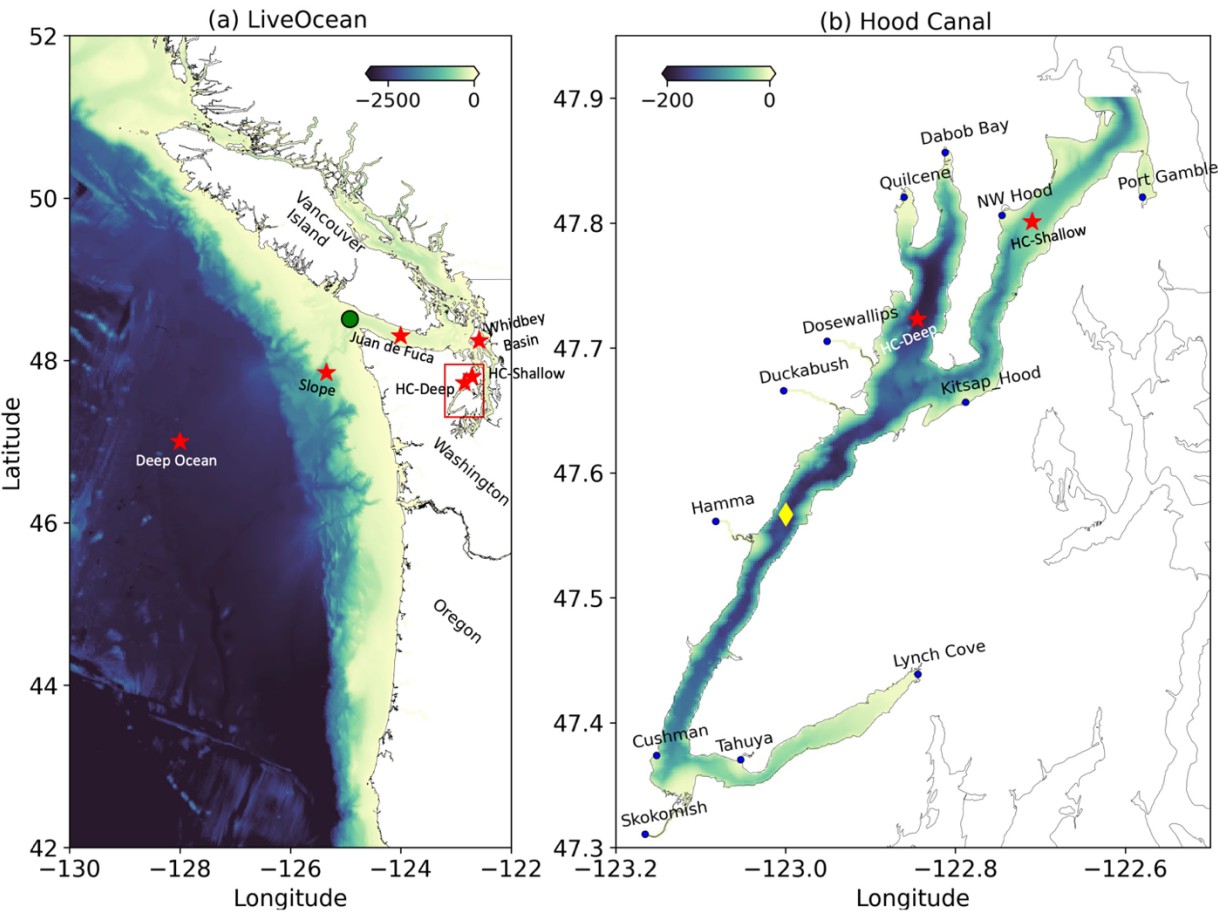

**Figure 1: (a) LiveOcean and (b) Hood Canal model domains and bathymetry. In (a), the red stars represent sites selected for 1-D vertical well mixed condition tests. The green dot indicates the particle release location to test offline particle tracking codes in the LiveOcean domain. In (b), the yellow diamond indicates particle and dye release location using the Hood Canal model domain. The blue dots represent locations with river inputs.**

## 2 Methods

### 2.1 Tracker

Tracker is an open-source Python-based Lagrangian particle packages, designed to work with ROMS hydrodynamic outputs. In addition to the Python standard library, other packages utilized include scipy (Virtanen et al., 2020), numpy (Harris et al., 2020), xarray (Hoyer and Hamman, 2017), and pandas (McKinney, 2010). Random displacement (as a modified random walk) is implemented in the vertical to represent the effects of turbulent mixing and prevent particles from unrealistically accumulating in low-diffusivity areas (Visser, 1997; North et al., 2006; Banas et al., 2009). The horizontal and vertical transport of particles are calculated as

$$x_{n+1} = x_n + u \cdot \Delta t, \tag{1}$$

$$y_{n+1} = y_n + v \cdot \Delta t, \tag{2}$$

$$z_{n+1} = z_n + (w + \frac{\partial AK_s}{\partial z}) \cdot \Delta t + R\sqrt{2AK_s \cdot \Delta t}, \tag{3}$$

where $x_n$, $y_n$, and $z_n$ are the horizontal and vertical particle positions (in meters) at time step $n$ after the advection, $\Delta t$ is the timestep, $R$ is a normally distributed random function with a mean of 0 and a standard deviation of 1. $AK_s$ is the vertical diffusivity evaluated at $(z_n + 0.5 \frac{\partial AK_s}{\partial z} \Delta t)$, and the derivative $\frac{\partial AK_s}{\partial z}$ is evaluated at $z_n$ (North et al., 2006). Before calculating the vertical derivatives, the vertical profile of eddy diffusivity $AK_s$ is smoothed using a 3-point Hanning window (Thomson and Emery, 2014) to reduce the potential sharp gradient in vertical diffusivity that could cause particle aggregations, following the 4-point and 8-point moving average used in North et al. (2006). Specifically, $AK_s[1:-1] = 0.25 \times AK_s[0:-2] + 0.5 \times AK_s[1:-1] + 0.25 \times AK_s[2:-1]$. The index follows rules in Python. In addition, the surface and bottom $AK_s$ are adjusted to be equal to the values one grid point in. The choice was motivated by the fact that we use a nearest neighbor search algorithm and were concerned that particles close to the top or bottom might use a near-zero diffusivity. A 4th-order Runge-Kutta integration scheme was used in Equation 1-3 to displace particle from the current location to the next location after an internal timestep of $\Delta t$. $\Delta t \ll min(1/AK_s'')$, where $AK_s''$ is the second derivative of the vertical diffusivity, is required to satisfy the vertical random displacement model criterion (Visser, 1997). The timestep for both horizontal and vertical particle tracking was set as 300 s after examining the vertical profiles of $AK_s''$ at six sites (used for well mixed condition test in section 2.1.1) from deep open ocean to the inner Salish Sea.

To speed computation, Tracker uses pre-computed nearest neighbor search trees to find velocities ($u$, $v$, $w$ in Equations 1-3) and other fields (e.g., diffusivity, temperature, and salinity) used for moving each particle forward. The accuracy of Lagrangian particle trajectory calculated with different numerical integrators and interpolation methods was discussed in Nordam and Duran (2020). In our development experiments, we found that the combination of nearest neighbor interpolation and 4th-order Runge-Kutta integrator can speed computation for the large grid size of the model domain and ensure the accuracy of particle trajectory in regions with complex shoreline geometries, e.g., the curving channels in Tacoma Narrow in the southern Salish Sea. The initial particle locations are seeded in the coordinate of longitude and latitude (an example can be found in lines 24-25 of experiments.py in the supplement). The horizontal advection (in meters) of particles are converted to degree using an Earth radius calculated based on the local latitude (earth_rad function was given in lines 250-264 of zfun.py in the supplement) and the particle locations (lon, lat) are saved in the format of NetCDF. For the land boundary, if a particle is advected onto land, it will be moved to a neighboring grid cell with a random direction. The numerical model does not resolve every process in the nearshore region (waves, rip currents, etc.), therefore, this is a practical way to make sure that particles do not get caught in the boundaries or in corners. To test if Tracker can give trustworthy results, one important test is the preservation of vertical well mixed conditions (North et al., 2006). Another is the similarity to dispersion of an inert dye.

**2.1.1 Well mixed condition test**

Using the hourly-saved hydrodynamic output from LiveOcean, six sites in different dynamic settings from the deep ocean to the Salish Sea (Figure 1a) were selected to perform the well mixed condition (WMC) tests on Tracker with horizontal and vertical advection turned off and a random displacement model implemented in the z direction (i.e., the particle location is only controlled by $z_{n+1} = z_n + \frac{\partial AK_s}{\partial z} \cdot \Delta t + R\sqrt{2AK_s \cdot \Delta t}$). For each site, 4,000 particles were seeded uniformly from the free surface to the bottom. The WMC tests were run for 12 hours with both a time-dependent diffusivity profile (from 2021.01.01 00:00:00 to 12:00:00) and a steady diffusivity profile (at 2021.01.01 00:00:00, Figure 2). The timestep for tracking particles in WMC tests is 300 s. To satisfy

the WMC test, the initially well-mixed particles are expected to remain uniform in a statistical sense regardless of the diffusivity profiles, in consistent with the Eulerian solution to the 1-D vertical diffusion equation ($\frac{\partial C}{\partial t} - \frac{\partial}{\partial z}\left(K\frac{\partial C}{\partial z}\right) = 0$, where $K$ is eddy diffusivity) with an initial uniform concentration, $C(z) = C_0$, and no flux boundaries, $K\frac{\partial C}{\partial z} = 0$ (Visser, 1997; Rowe et al., 2016; Nordam et al., 2019). Metrics of success for WMC tests follow North et al. (2006) that particle numbers were compared to a "non-significant range" to test whether the WMC was satisfied. To obtain the non-significant range (dash lines in Figure 2), 4,000 snapshots of 4,000 randomly distributed particles were generated and the number of particles was then calculated in 28 evenly spaced intervals. The mean values of the highest (187.3) and lowest (102.5) value of particle numbers in each interval from the 4,000 snapshots were used to define the upper and lower limit of the non-significant range (North et al., 2006).

## 2.2 Other offline particle tracking software packages

Here we briefly describe the three other offline particle tracking packages: LTRANS (North et al., 2006), Particulator (Banas et al., 2009), and OpenDrift (Dagestad et al., 2018), with more details about their configurations given in Table 1 and provided in respective references.

LTRANS is a well-documented tool written in Fortran 90, specifically for output from ROMS. It has broad applications in studying larvae transport (North et al., 2008), oil spills (North et al., 2011; Testa et al., 2016), coastal connectivity (Li et al., 2014), plastics (Liang et al., 2021), algae (Wang et al., 2022), etc. Particulator is written in MATLAB, mostly specific to output from ROMS, and has been used to study water pathways (Banas et al., 2015; Stone et al., 2018), and harmful algal bloom (Giddings et al., 2014). OpenDrift is written in Python and has flexibility to work with forcing data from different ocean models, including ROMS. It has rather wide-ranging applications in tracking particles with diverse properties, e.g., fish eggs (Melsom et al., 2022), Environmental DNA (Andruszkiewicz et al., 2019), oil, chemical tracers, sediment, capsized boats, icebergs, etc. (Dagestad et al., 2018). Given the different interpolation schemes, numerical integrators, and how turbulent dispersion and encounters with model boundaries are treated, we limit our inter-model comparisons by only considering advection of passive (or neutrally buoyant) particles by the three-dimensional flow and vertical turbulent mixing (without surface windage and waves).

## 2.3 Online passive dye experiment and particle tracking

A passive dye experiment was conducted to determine if the particle-tracking model predictions agree with simulated diffusion. Dye can be considered the "truth" that particle tracking codes seek to replicate. However, this idea is complicated by the presence of numerical mixing which is intrinsic to model advection algorithms (Burchard and Rennau, 2008; Ralston et al., 2017). Numerical mixing, defined by the decrease in tracer variance due to discretization errors in the tracer advection scheme, increases the dispersion of tracers (Ralston et al., 2017). In Broatch and MacCready (2022), numerical mixing was found to account for one-third of the total mixing of salinity in the LiveOcean Model inside the Salish Sea. While most model studies do not quantify numerical mixing, those that have, mostly limited to estuaries, show that it is significant. Thus, we expect in general that dye will experience greater horizontal and vertical dispersion than particles, especially in regions with strong horizontal gradients.

Using the Hood Canal model, a passive dye was introduced from a grid cell in the middle of water column of the channel (Figure 1b) and was tracked for 7 days starting from 2021.06.01 00:00:00. Before activating the dye module, the hydrodynamic simulations were run for the whole year of 2021 with daily saved restart files. An additional variable 'dye_01' was added to the restart file at 2021.06.01 00:00:00 with a concentration of 1 in the selected grid cell and 0 elsewhere. The timestep for dye transport is 40 s. The

MPDATA advection scheme (Smolarkiewicz, 1984) was applied for dye, the same as temperature and salinity. This scheme effectively reduces numerical dispersion and prevent negative concentration values (Melsom et al., 2022).

To compare Eulerian dye and Largangian particles, $10^5$ particles with a distribution of $100 \times 100 \times 10$ (longitude, latitude, vertical) were released from the same model grid cell at the same time as dye release for all four offline tracking codes. Particle tracking was driven by the hourly saved history files in each case. Each particle was associated with a particular mass $\varepsilon_0$ obtained as the ratio of the initial dye mass to the total particle number (i.e., $10^5$). The timestep for offline particle tracking is 300 s. Previous experiments with Tracker showed this time step was required in LiveOcean in regions with strong currents and complex channel shape. Longer time steps would sometimes advect particles over narrow land regions instead of following curving channels. A slightly different seeding strategy was applied for ROMS online particle module for convenience. The $10^5$ particles were distributed uniformly along the diagonal of the selected model grid cell, which gives the same initial centers of mass for particles in x, y, and z dimensions. The thickness of the selected model grid cell is about 5% of the total local depth, and the adjusted particle initialization in ROMS online tracking is expected not to significantly influence the intercomparisons. The timestep for online tracking is 40 s. Additional comparisons for the four offline particle packages were conducted using the large LiveOcean model domain and its hourly saved history file. $10^4$ particles were evenly distributed within a 1 km × 1 km square at the free surface and in the middle water column near the mouth of the strait of Juan de Fuca (Figure 1a). Particles were tracked for 7 days from 2021.01.01 00:00:00 with a timestep of 300 s. In all experiments mentioned above, dye concentration and particle trajectory positions were saved hourly for further analysis.

To compare the mean pathways of dye and particles, their centers of mass were calculated as

$$M_{x\_dye}(t) = \frac{\sum_{i=1}^{N_{total\_grid}} x_{i\_dye} \cdot C_{i\_dye} \cdot V_i}{\sum_i C_{i\_dye} \cdot V_i}, \tag{4}$$

$$M_{x\_particle}(t) = \frac{\sum_{i=1}^{N_{total\_particle}} x_{i\_particle}}{N_{total\_particle}}, \tag{5}$$

where $N_{total\_grid}$ is the total number of model grid cells, $N_{total\_particle}$ is the total particle number, $C_{i\_dye}$ is the dye concentration in model grid cell $i$, and $V_i$ is the corresponding grid cell volume. The centers of mass in y and z dimensions were calculated with similar equations.

## 3 Results & Discussion

### 3.1 Well mixed condition tests

The vertical particle distributions from WMC tests for Tracker are shown in Figure 2. Results at other locations (not shown) gave similar results. The site with deeper depth passed WMC tests for both time-dependent and steady vertical diffusivity profiles. Occasional failures of WMC tests were found at the shallow site, specifically the vertical regions with low diffusivity and increasing gradient. Particles tend to cluster in low diffusivity regions, e.g., ~38 m in the site HC-shallow (Figure 2b, 2f). Previous studies suggested that demonstration of WMC was influenced by discontinuities in $AK_s$ profiles, the interpolation scheme used to estimate $AK_s$ and its vertical gradient, and the timestep of particle tracking (Brickman and Smith, 2002; North et al., 2006). Here we demonstrated that the 3-point Hanning window used to smooth $AK_s$ profiles, the nearest neighbor interpolation scheme used to obtain $AK_s$, and a timestep of 300 s in Tracker generally passes the WMC test for sites from offshore deeper than 2,500 m to the Salish sea shallower than 40 m, however there are occasional failures. We proceed by assuming that the effects of such failures would in practice be smeared out as particles are moved rapidly by tidal advection through a wide range of conditions.

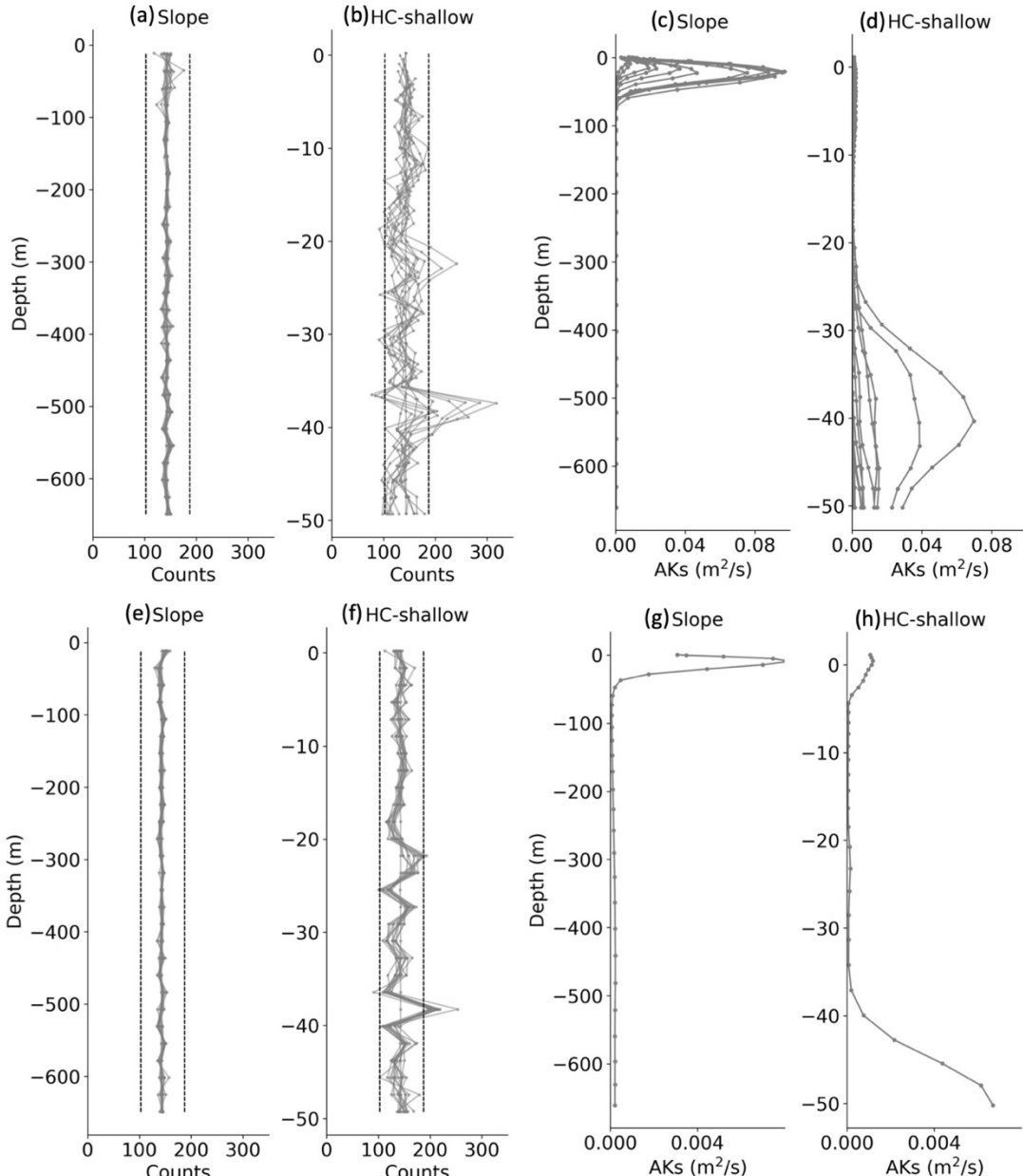

**Figure 2: 1-D vertical well mixed condition (WMC) tests at two sites (Figure 1a) in LiveOcean model domain and the associated profiles of vertical diffusivity (c-d, g-h). (a-b) WMC tests using time-dependent diffusivity profiles shown in (c-d). (e-f) WMC tests using steady diffusivity profiles in (g-h). All WMC tests were conducted for 12 hours with hourly output and a timestep of 300 s. The dashed lines in (a-b, e-f) indicate the non-significant range, outside which the WMC tests fail.**

**3.2 Comparisons among particle tracking software packages using the Hood Canal model**

**3.2.1 Offline versus online particle tracking**

Centers of mass of trajectories from all particle tracking codes, relative to the initial release location, are shown in Figure 3. Particles were initialized at the low tide and the tracking was followed by a flood tidal phase. A relatively good inter-model match was achieved for the first 5-6 hours during the flood tide. After this point, all models still tend to follow the same trend, but drift apart presumably because of these different interpolation and advection schemes and online or offline tracking. The differences increase with time since different particle locations sample different velocities and diffusivities.

Horizontal spreading of vertically integrated particle mass (Figure 4) and vertical distributions of particles (Figure 5) exhibit similar but not equivalent evolutions among all tracking codes. Particles from OpenDrift tend to be less spreading. Generally, results from online tracking stays in the middle of other offline tracking codes. Compared to offline tracking, online particle trajectory is updated every timestep along with the hydrodynamic model runs and vertical transport is better accounted (Ricker and Stanev, 2020). However, offline tracking provides more flexibility to incorporate forcings from more than one numerical model or observational databases. In offline mode, it is easier to modify algorithms to include user-defined processes (e.g., diel vertical migrations, settling and resuspension) and test parameters or different particle seeding strategies without rerunning the full ocean model, which can be computationally expensive (Dagestad et al., 2018; Hunter et al., 2022; Melsom et al., 2022). Simulation backward in time is also more easily performed offline. To the best of our knowledge, no studies so far have targeted backward tracking using online particle tracking models. On the other hand, updating trajectories in offline tracking could suffer from inaccuracies induced by interpolation scheme since it reads subsampled or averaged model outputs, which could smear out short-time and small-scale advective processes simulated by ocean circulation models (Wagner et al., 2019; Melsom et al., 2022).

**3.2.2 Lagrangian particle tracking versus Eulerian passive dye**

Using the Eulerian dye model prediction as a benchmark, we evaluate the performance of Lagrangian particle tracking models. Like the comparisons among different particle tracking codes, the particle and dye models also agree well with each other within the first few hours following their initial release (Figure 3). The evolution of the center of mass from online particle tracking matches the best with the center of mass of dye. The horizontal center of mass of dye stays between all particle tracking models, while dye predicts somewhat deeper mixing than particle models, with the vertical center of mass being about 5-10 m deeper after 30 hours (Figures 3c). Greater vertical spreading of dye was also observed in the histogram (Figure 5). Dye fills the upper 20-140 m after 2 days while particles are still confined to a depth range of 50-90 m around their release depth. To obtain the histogram of vertical dye distribution, dye mass inside each model grid cell was converted to an equivalent number of particles via the constant $\varepsilon_0$ (defined in section 2.3). The vertical coordinate in the center of the grid cell that contains dye was then used to represent the vertical location of dye-converted particle number. This conversion might lead to the spiky vertical distribution in the early stage of dye transport as seen in Figure 5a.

The horizontal spread of vertically integrated dye and particle mass is shown in Figure 4. Generally, dye is also more widespread than particles in the horizontal (similar to patterns observed in e.g., Melsom et al., 2022; Nepstad et al., 2022). Low values of dye spread faster than particles and cover a greater area. However, the spread of high mass concentration exhibits a reasonable degree of similarity, indicating that Lagrangian particle tracking models all yield similar simulations of vertical dispersion, although formulations for particles and dye transport differ largely in details (North et al., 2006). It is suggested (North et al., 2006; Wagner et al., 2019; Broatch and MacCready, 2022; Nepstad et al., 2022) that the vertically reconstituted diffusivity profile, vertical model

grid resolution, total particle number, the temporal subsampling of velocity fields, and numerical mixing influencing dye
concentration give rise to the deviations between particle tracking and inert dye component.

Comparing online particle tracking and online passive dye experiment in ROMS, Lagrangian particle tracking tends to be more computationally demanding (Table 2). The average time for running hydrodynamics for one day in Hood Canal model is about 160 s using 200 cores on a Linux cluster. The running time increases a little to 196 s with dye module activated, and it increases to 1218 s when floats module was activated to track $10^5$ particles. However, particle tracking, especially offline tracking, is more
flexible, and dye calculations can be more costly in some instances. For example, multiple passive dyes are required to represent multi-component river-borne discharges (e.g., nutrients, pathogens, freshwater) but particles can carry all these properties in one trajectory tracking experiment (Banas et al., 2015). Particle tracking is also economical in disk space since only particle locations and associated water properties, e.g., salinity and temperature, are stored but dye is usually saved for the whole model domain (Melsom et al., 2022). In addition, particle tracking models can resolve particle displacement at sub-grid scales (Alosairi et al.,
2020; Xiong et al., 2023) because dye is a grid cell property.

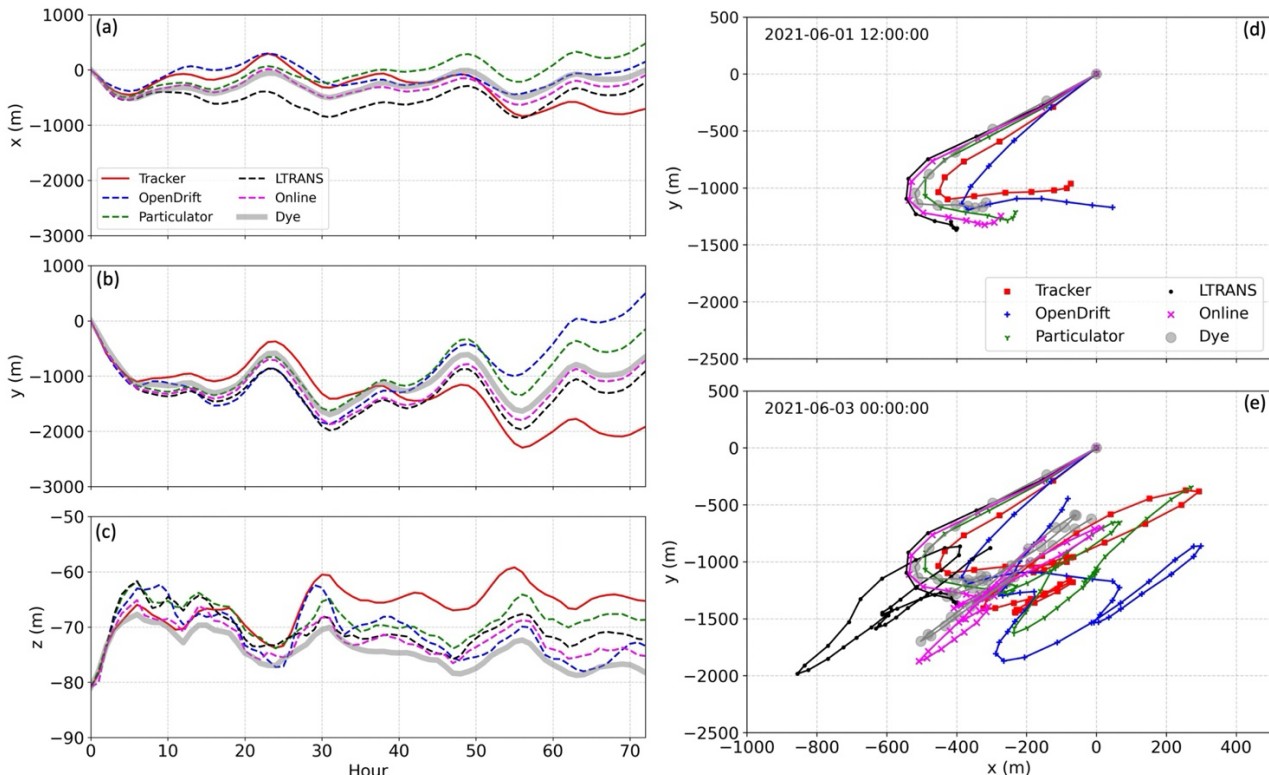

**Figure 3: The centers of mass in x, y, z directions obtained from offline and online particle tracking and passive dye experiments using Hood Canal model. (a-c) evolution of the centers of mass. (d-e) the centers of mass in x and y directions with particles tracked for (d) 12 hours and (e) 48 hours. Particles and dye were released inside Hood Canal at 2021.06.01 00:00:00 (Figure 1b).**

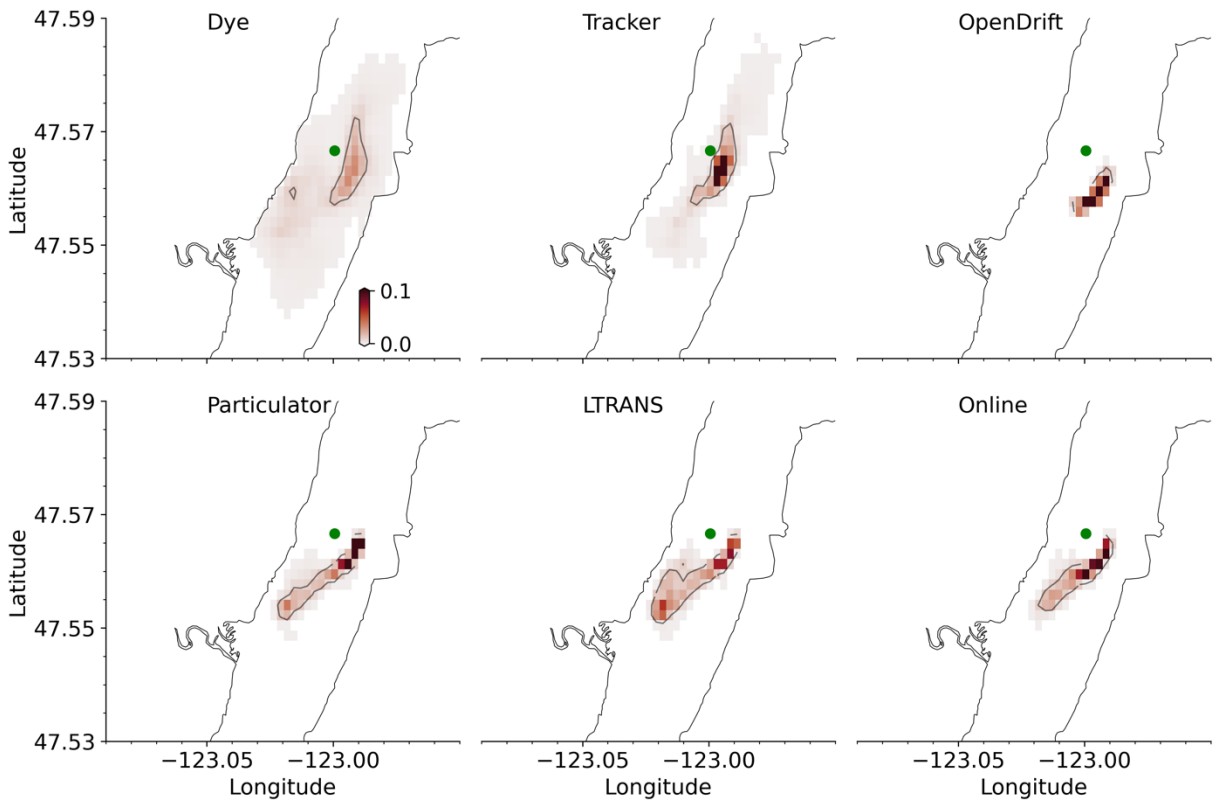

**Figure 4: Snapshot of vertically integrated dye and particle mass (scaled to 0-1 by the initial dye or particle mass) after 1-day of simulation using the Hood Canal model. The green dot indicates the initial dye and particle release location. The black contour in each panel represents a value of 0.01.**

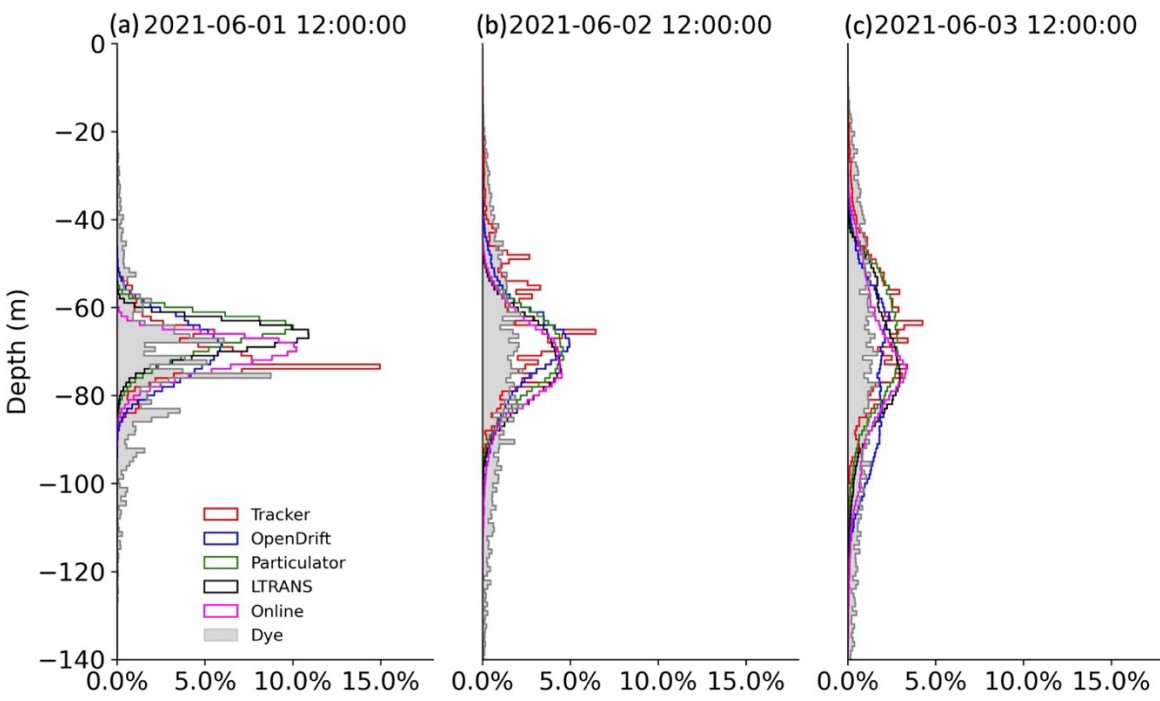

**Figure 5: Histogram of vertical particle and dye distributions using Hood Canal model at 12, 36, and 60 hours after release. The effects of numerical mixing may be the cause of the greater vertical spread of dye vs. particles.**

### 3.3 Comparison among offline particle tracking software packages using the LiveOcean model

Additional comparisons just among the four offline particle tracking codes were conducted using the larger LiveOcean model domain. Particles were released from the free surface and the middle water column in the coastal area at 2021.01.01 00:00:00 (Figure 1a) and particle dispersal regions along the coast are with a horizontal grid resolution of ~1000 m (Figure 6d, h). The particle tracking period was dominated by southerly winds, favorable for northward and onshore near-surface currents over the shelf (Giddings et al., 2014). Thus, particles exhibit net northward transport, and the surface-released particles move closer to the

coast (Figure 6h-i). Besides the center of mass, particle density, a ratio of the vertically integrated particle numbers in each horizontal grid cell to the respective grid cell area, was also calculated (Figure 7). A relatively good match in the center of mass among these tracking codes is evident for about 1 day of tracking (Figure 6). This suggests that the decorrelation time (Klocker and Abernathey, 2014) is about 1 day in this region. After that, the centers of mass still follows a similar trend but with increasing separations. Note that Tracker produced a large vertical downward displacement during hours 13-22 (Figure 6g) in the case of

mid-depth release, likely due to the greater vertical velocity and weaker stratification experienced by the center of particle mass. The horizontal center of mass calculated by Tracker is closer to the coastline within this period (Figure 6h). Generally, the horizontal advection due to different interpolation and integration methods leads particles to different dynamic environments and results in greater (or less) vertical advection. The spatial coverages of particles in the horizontal and vertical also share similar patterns but exhibit somewhat different local accumulation patches (Figures 7-8). As we saw in the Hood Canal experiments, all

four offline particle tracking codes have similar performance when they track the same water mass in the coarser model domain and in the shelf environment. Note, however in Figure 8 that there are real differences in the details of vertical particle distribution among the models after 2 days. These result in part from the details of the algorithms used for vertical dispersion, and in part from particles experiencing different vertical mixing associated with different horizontal locations.

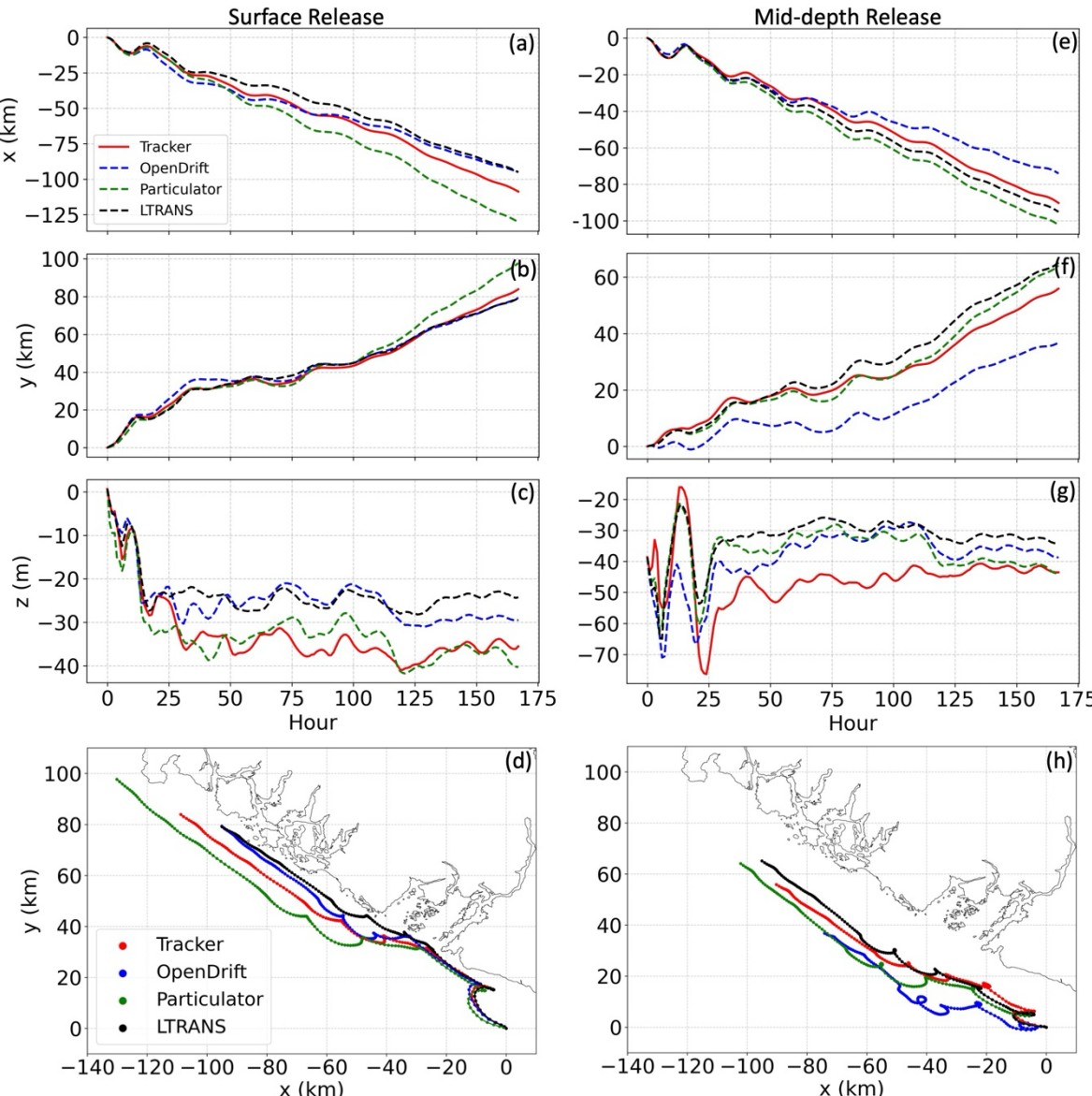

Figure 6: The centers of mass in x, y, and z directions for all four offline particle tracking codes simulated using hydrodynamic outputs from the LiveOcean model. (a-c) particles released from the free surface. (e-g) particles released from the middle water column. (d, h) centers of mass in x and y directions.

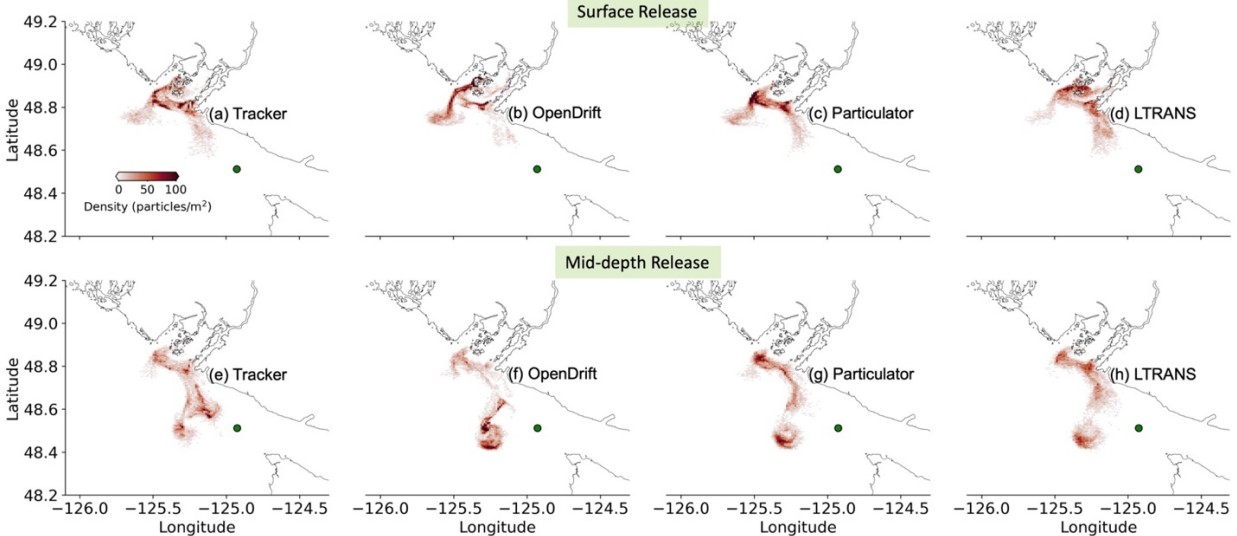

**Figure 7: Vertical integral of particle densities for all four offline particle codes after 2 days of tracking using hydrodynamic outputs from the LiveOcean model. (a-d) particles were released from the free surface, (e-h) particles were released from the middle water column. Green dots represent particles release location.**

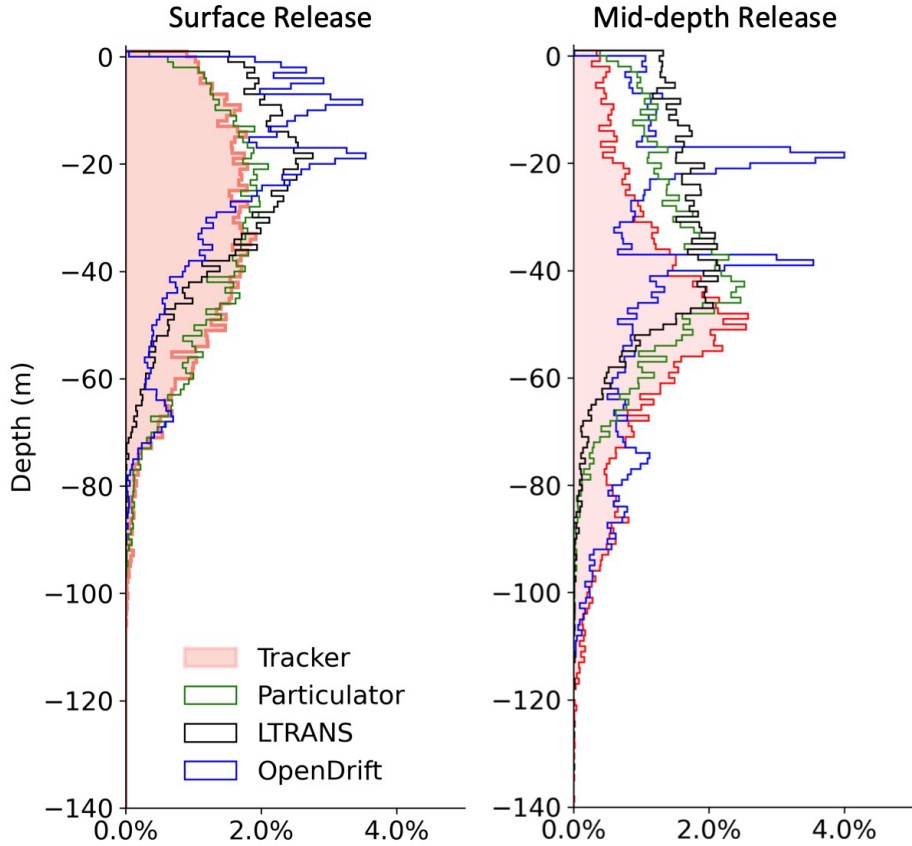

**Figure 8: Histogram of vertical particle distribution for all four offline particle codes after 2 days of tracking using hydrodynamic outputs from the LiveOcean model. Left panel: particles released from the free surface; right panel: particles released from the middle water column.**

### 3.4 Computation time

Although all tested offline particle tracking codes share similar predictions compared with online particle tracking and online Eulerian dye, especially for the first few days, computation efficiency is another important metric for their performance evaluation. The computation time for each offline tracking code was recorded using both the small domain of the Hood Canal model and the large domain of the LiveOcean model (Figure 9; Table 3). Each particle model was run to track neutrally buoyant particles for 25 hours with a timestep of 300 s. Particle locations and temperature and salinity at each particle's location were saved hourly. The total particle number varied from $10^0$ to $10^6$. The computation time tests were conducted using an Apple M1 Pro for Particulator and OpenDrift, and a Linux machine for LTRANS. We recorded the computation time of Tracker both on the Apple M1 Pro and the Linux machine.

The two tracking codes, Tracker and OpenDrift, that were written in Python, have close computation costs when Tacker was tested on the Linux machine and OpenDrift was tested on a laptop (Figure 9), while the performance of Tracker on the same laptop is faster by a factor of 2-3 compared to that on the Linux machine, perhaps because of different file access speeds between solid-state and RAID drives used to store the model output. The computation time of Tracker and OpenDrift increases with increased particle number, with the largest increase when running one million particles. The LTRANS, written in Fortran, runs fast with a small number of particles but the computation requires a much longer time than other codes with increased particle number and even

becomes prohibitive when tracking one million particles in the large LiveOcean domain with a grid dimension of 1302×662× 30. Generally, more time is required to track particles in a larger model domain than a smaller one for all offline tracking codes. One interesting finding is that for Particulator, written in MATLAB, the computation time is only weakly influenced by the total particle number and the code can run very fast with the one million particles. In the large LiveOcean domain, generally, Tracker requires the least computation time among those tracking packages that were tested on the same laptop, for example, Tracker is 10 times faster than Particulator when the tracking number is less than $10^4$. The interpolation and advection scheme, algorithm structures, and programming languages could all affect the computation cost (Table 1). It is beyond the scope of this paper to explore the detailed tradeoffs between these factors, and instead hope the results of computation time may be one piece of information scientists can use when choosing a particle tracking package and designing an experiment.

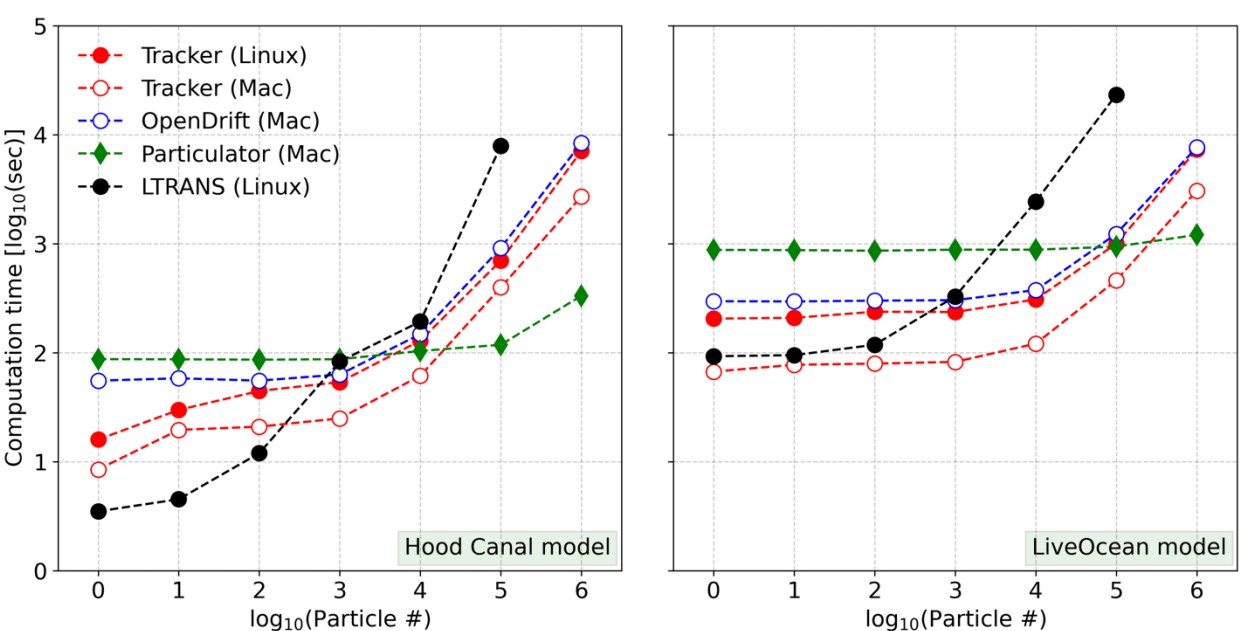

**Figure 9: Computation time for tracking particles for 25 hours with a timestep of 300s for all four offline particle tracking codes. The total particle number increases from $10^0$ to $10^6$. The computation time for LTRANS was obtained on a Linux machine, while the computation time for Particulator and OpenDrift was obtained on Apple M1 Pro. Tracker was tested both on Linux machine and Mac. The computation cost for LTRANS with 1 million particles is prohibitive on the Linux machine when testing it with the large LiveOcean domain; for the small Hood Canal domain, the computation time of LTRANS is about 25 hours estimated from the timestamp of the hourly output files that were saved separately.**

## 4. Conclusions

In this work, we introduced a new offline Lagrangian particle tracking model, Tracker v1.1, and tested its ability to preserve the vertical well mixed conditions. We also evaluated its performance compared with online Eulerian dye, one online and three offline particle tracking codes using a high-resolution (200 m grid size) ocean circulation model. Additional comparisons were performed for all four offline tracking codes in a larger model domain with a horizontal grid resolution of ~1000 m in the particle tracking region.

We show that the mean advection pathways and spatial distributions of dye and particles are reasonably similar when they were tracking the same water mass. The spreading of Eulerian dye is more dispersive with a wider distribution of low concentrations. Similar inter-model comparisons were observed in both small (fine) and large (coarser) model domains. The passive dye was solved in a fixed Eulerian framework that addresses the advection and diffusion equation which might suffer from spurious numerical mixing. The Lagrangian particle tracking model employs a movable frame of reference. Online tracking may be expected to give more accurate results because it uses a much shorter time step between velocity fields but lacks flexibility compared to offline tracking. In our experiments, results from online particle tracking were not obviously different from that of any of the offline tracking packages. Although offline tracking is influenced by subsampled model output, parameterization of vertical turbulence mixing, and interpretation scheme, its flexibility and reliability against passive Eulerian dye and online tracking make it a useful and cost-effective tool in tracking transport pathways in oceanography. Finally, the reasonable preservation of well-mixed conditions, speed improvements in large model domains, and similar performance against other particle tracking codes and passive dye achieved by Tracker suggest that it is a reliable and efficient particle tracking package to use with ROMS. All tests in this study used a ROMS grid aligned along lines of constant latitude and longitude. In principle, Tracker should work on a more general grid, but this has not been tested.

## Code/Data availability

All example codes and hydrodynamic inputs that were used to test these five particle tracking packages are available at https://doi.org/10.5281/zenodo.10810102. The original source codes for all offline tracking packages can be found: Tracker (https://github.com/parkermac/LO/tree/v1.1/tracker); OpenDrift (https://opendrift.github.io/); LTRANS (https://northweb.hpl.umces.edu/LTRANS.htm); Particulator (https://github.com/neilbanas/particulator).

## Author contribution

PM is the primary developer for Tracker and the LiveOcean model. JX developed the Hood Canal model and carried out all the particle tracking and dye release experiments. JX drafted the manuscript and PM contributed to the review and editing.

## Competing interests

The authors declare that they have no conflict of interest.

**Acknowledgments**

This study is based upon research supported by the Office of Naval Research under Award Number (N00014-22-1-2719) and
supported by NSF Award 2122420 and NOAA grant NA21NOS0120168-T1-01. We would like to thank David Darr for technical
support, Elizabeth Allan for sharing example codes to run OpenDrift, Elias Hunter and Elizabeth North for useful suggestions to
run LTRANS, Aurora Leeson for preparing the tiny river discharge data that were included into Hood Canal model domain, Erin
Broatch for processing the ORCA buoy data for Hood Canal model calibration, and Dongxiao Yin for useful discussions during
the preparation of this manuscript. We also appreciate one anonymous reviewer, Dr. Tor Nordam, and topic editor, Dr. Ignacio
Pisso for their constructive suggestions and comments on the manuscript.

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

Table 1. Configurations for different particle tracking codes.

| | Tracker | OpenDrift | Particulator | LTRANS | ROMS online floats |
|---|---|---|---|---|---|
| Programming language | Python | Python | MATLAB | Fortran | Fortran |
| Time step | Self-defined | Self-defined | Self-defined | Self-defined | Same as baroclinic time step |
| Vertical turbulence | Random displacement | Random displacement | Random displacement | Random displacement | Random displacement |
| Land boundary | Move particles on land to the middle of the nearest wet rho point | • Stranding: particles deactivated<br>• Previous: particles moving back to previous locations<br>• None: particles not interacting with land | By default, particles carried within one grid cell of land will wait there until flow can carry them away. | Particles reflected off the land boundary with an angle the same as the approach angle. The reflect distance equals the distance that particles exceeded the boundary. | Particles avoid crossing land boundary |
| Open boundary | Not specified, remove particles outside boundary in post-processing | Particles deactivated outside the domain (or absorbing boundary) | Particle stops if it will be transported outside the domain in the next timestep and wait there until flow moves it around inside the domain | • Reflective boundary: treated the same way as land boundary.<br>• Sticking boundary: stop moving | Particles outside open boundary are deactivated |
| Vertical boundary | Reflect vertically back into the domain by a distance that the particle exceeds the boundary or enforce limits on reflection with the numpy remainder function if the vertical advection moves particles more than the total water depth | • Bottom boundary: lift_to_seafloor, deactivate, previous, or resuspended<br>• Surface boundary: reflective or stick to surface | Particles move outside the surface (or bottom) will be put back in sigma = 0 (or -1) | Reflect vertically back to the domain with the same distance that particles exceed the boundary | • ifdef float_sticky: floats that hit the surface are reflected; floats that hit the bottom get stuck;<br>• undef float_sticky: floats that hit the surface or bottom are reflected |
| Advection scheme | 4th-order Runge-Kutta | Euler, 2nd-order Runge-Kutta, 4th-order Runge-Kutta | 2nd-order Runge-Kutta | 4th-order Runge-Kutta | 4th order Milne predictor and 4th order Hamming corrector |
| Interpolation scheme | Nearest neighbor | Bilinear | Bilinear | Water-column profile scheme for 3D and bilinear for 2D variables | • Inside masked cells: linear & nearest neighbor<br>• Outside masked cells: bilinear |
| Backward tracking | Able to include | Yes | Able to include | Yes | Not able to do backtracking |
| Ease of use | • Read ROMS history file<br>• No compilation required, easy to set up python environment<br>• Flexible to define the initial particle release location and add user-defined functions<br>• Running platform independent | • Needs to concatenate grid information to ROMS history file<br>• No compilation required, easy to set up python environment<br>• Flexible to define the initial particle release location, modify existing modules, and write user-defined modules.<br>• Running platform independent | • Read ROMS history file<br>• No compilation required but MATLAB is a commercial software<br>• Flexible to define the initial particle release location<br>• Running platform independent | • Read ROMS history file but each file must have at least 3 timesteps<br>• Take time to compile source code<br>• Flexible to define the initial particle release location<br>• Run on Linux machines<br>• Require a long time run for large number of particles | • Require experience to compile ROMS source code and set up HPC environment<br>• The initial particle release location seems to be not very handy/flexible to specify<br>• Run on Linux machines in parallel mode |
| Source code | https://github.com/parkermac/LO/tree/v1.1/tracker | https://opendrift.github.io/ | https://github.com/neilbanas/particulator | https://northweb.hpl.umces.edu/LTRANS.htm | https://www.myroms.org/wiki/floats.in |

Table 2. Computation time (second) for ROMS simulations conducted for 1 day.

| Cases | Average* computation time for 1-day run |
|---|---|
| Hydrodynamic run | 160 |
| Hydrodynamic run + passive dye | 196 |
| Hydrodynamic run + online particle tracking (100,000 particles) | 1218 |

*Averaged from 2021.06.01 to 2021.06.07

** All cases were run on UW's Hyak supercomputer with 200 cores.

Table 3. Computation time (second) for different particle tracking software packages using the Linux machine or Mac with hydrodynamic outputs saved from a small Hood Canal model domain and a large LiveOcean model domain. All cases were run for 25 hours with a timestep of 300 s and hourly output with particle locations, and temperature/salinity recorded by each particle.

| Particle number | Hood Canal model | | | | | LiveOcean model | | | | |
|---|---|---|---|---|---|---|---|---|---|---|
| | Tracker (Linux) | Tracker (Mac) | OpenDrift (Mac) | Particulator (Mac) | LTRANS (Linux) | Tracker (Linux) | Tracker (Mac) | OpenDrift (Mac) | Particulator (Mac) | LTRANS (Linux) |
| 1 | 16.0 | 8.5 | 55.5 | 87.4 | 3.5 | 205.8 | 67.2 | 297.2 | 878.8 | 92.9 |
| 10 | 29.9 | 19.6 | 58.3 | 87.1 | 4.5 | 209.2 | 77.6 | 295.8 | 875.0 | 95.1 |
| 100 | 44.6 | 21.0 | 55.5 | 86.4 | 12.0 | 237.9 | 79.5 | 300.7 | 863.3 | 118.4 |
| 1,000 | 53.8 | 25.0 | 63.2 | 87.3 | 83.2 | 236.6 | 82.5 | 303.8 | 883.0 | 328.7 |
| 10,000 | 129.1 | 61.5 | 148.2 | 103.9 | 193.5 | 308.9 | 121.3 | 376.1 | 882.7 | 2437.3 |
| 100,000 | 701.0 | 398.0 | 915.7 | 118.4 | 7931.8 | 1001.9 | 462.0 | 1234.0 | 941.8 | 23325.7 |
| 1000,000 | 7093.0 | 2715.2 | 8419.8 | 333.0 | / | 7309.3 | 3063.7 | 7678.2 | 1208.0 | / |