# Peer review of "Intercomparisons of five ocean particle tracking software packages in the Regional Ocean Modeling System"

_Geoscientific Model Development, 2023_

## Author Comment (AC3)

**GMD-2023-45 Response to Reviewers**

We appreciate the time and efforts one anonymous reviewer and Dr. Tor Nordam dedicated to providing feedback on our manuscript. We have incorporated or responded to all the comments and suggestions made by the reviewers. Please see below, in blue, for a point-by-point response to the reviewers' comments and concerns. The line numbers refer to the tracked version of our manuscript.

**Reviewer 1:**

Review of "Intercomparisons of five ocean particle tracking software packages" by Xiong and MacCready

In this manuscript, the authors present a new particle tracking code and compare it to three existing offline codes, one online code, and an online dye simulation. They focus on two regional domains on the west coast of the US. They show that the new code compares well in terms of accuracy and efficiency.

The manuscript is generally well-written. However, I have some significant concerns that prevent me from recommending its publication in GMD. Most importantly:

1. It is unclear what this new Lagrangian tracking code adds. What is the advantage of the new code over the previous codes? What does it add/improve on the other codes? That could be much more explicit.

Response: we thank the reviewer for the suggestion and apologize for the lack of clarity. In this study, we introduced a new offline particle tracking code Tracker and evaluated its performance with other offline and online tracking codes and passive dye. The main purpose is to conduct the intercomparisons of some commonly used codes in the same numerical simulations to explore the net effect of the many slightly different choices made by the different developers. The other offline tracking codes have already been rigorously tested by their developers, and we present our own tests of vertical mixing for Tracker. When choosing a particle tracking code to use, modelers have many considerations. Will the code be easy to use with their model output? Will they be able to modify the code for their specific needs, e.g., introducing vertical behavior? Will it run fast enough? Finally, a modeler should have some confidence that regardless of which code they choose the results will be reasonably similar for all the choices. The goal of this intercomparison is primarily to address this final issue of confidence, and to discuss some of the other tradeoffs. We have endeavored to clarify this in the revised manuscript. We now explicitly describe our goal in Introduction (lines 93-100) and also add more details on the implementation of Tracker in Methods (lines 113-152).

2. The title is far too general, and suggests a much wider scope than that the manuscript can deliver. It's therefore not appropriate for this specific manuscript.

Response: thanks for the suggestion. We revised the title to "Intercomparisons of five ocean particle tracking software packages in the Regional Ocean Modeling System" to be more specific.

3. It's not at all clear why the LiveOcean and Hood Canal are such good testcases for Lagrangian models. I would have expected a much more thorough discussion of why these are specifically suited. Now, it seems as if the authors had these models lying around and decided to do the comparison; instead of selecting an optimal case for the comparison.

Response: the motivation of the present study for particle tracking code intercomparison stems from the authors' research need to decide which particle code to use for our own ROMS model analysis. During our experiments, we found several open-accessed particle tracking codes could be used but some code like Parcels required re-gridding of the original velocity fields, which requires extra work and will likely introduce some errors in re-gridding. Thus, we limited our comparisons among those codes that can directly operate on ROMS native velocity outputs. As suggested above, we revised the title to be specific to the ROMS model. Given that there is a considerable user group for ROMS and particle tracking is a very useful tool in oceanography, we thought our work of intercomparison could be useful to some part of the research community.

The LiveOcean domain includes deep open ocean, continental slope and shelf, and an inland fjord-type estuary with dynamic sills and quiet deep basins, providing diverse environments to test the preservation of the vertical well mixed condition. The saved LiveOcean model output database (2017-present) is convenient to test the offline particle codes but to do online particle tracking and dye experiment, we found it is more practical to implement them in a small model domain (will be explained more in comment 8).

The reviewer is correct in surmising that we "had these models lying around" and this clearly influenced our choice of experiments. While we did our best to select times and places in the models that we felt covered a useful range of parameter space for the coastal and fjordal ocean, there are clearly many cases we did not test, for example shallow intertidal areas with wetting and drying, river plume fronts, and so on. The "optimal case" for such a comparison is likely to be different for different modelers, hence the definition of globally optimal test cases would involve a much larger number of experiments than we could undertake.

4. It's unclear why some of the choices for the tracker code have been made. E.g., why does it employ nearest neighbour interpolation? That is not very customary for Lagrangian codes. And why then also use 4th order Runge-Kutta integration? Why aim for such high accuracy in time, when spatial accuracy is low?

Response: during the development of Tracker, we tested nearest neighbor and bilinear interpolation and found these two methods gave very similar results but nearest neighbor speeds up the computation in our large model grids. We also tested $2^{nd}$-order Runge-Kutta integration and found that a higher-order integrator is required to move particles forward in regions with complex shoreline geometries, like the curving channels in the Tacoma Narrow (in the southern Salish Sea). We justified our choice of nearest neighbor interpolation and $4^{th}$-order Runge-Kutta integration in lines 140-144.

5. The argument for smoothing the AK_s diffusivity field is unclear; what is the advantage of this?

Response: we added the argument for smoothing the vertical diffusivity in lines 132-133. Smoothing the vertical diffusivity can reduce the potential sharp gradients in vertical diffusivity that could cause particle aggregations (North et al., 2006).

North, E. W., Hood, R. R., Chao, S. Y., Sanford, L. P.: Using a random displacement model to simulate turbulent particle motion in a baroclinic frontal zone: A new implementation scheme and model performance tests. J. Mar. Syst., 60(3-4), 365-380, https://doi.org/10.1016/j.jmarsys.2005.08.003, 2006.

6. The discussion of the Well mixed condition test in section 2.1.1 could be more elaborate. What is the equation that is tested. Why? How is e.g. the non-significant range defined in Figure 2?

Response: thanks for the suggestion. We added more descriptions about the vertical well mixed condition tests in lines 157-186.

7. One of the most difficulty things to do for Lagrangian codes is boundary conditions near land, and avoiding stuck particles. While the strategies of each code is listed in table 1, there is no discussion of how well the tracker code performs near boundaries. This would be important information for potential users, especially in domains like the Hood Canal where there is so complicated topography.

Response: in Tracker, if a particle gets onto land, it will be moved to a neighboring grid cell with a random direction (please see line 471 in https://github.com/parkermac/LO/blob/v1.1/tracker/trackfun.py). The numerical model does not resolve every process in the nearshore region (waves, rip currents, and so on), therefore, this is a practical way to make sure that particles do not get caught in the boundaries or in corners. We added these details in lines 149-152.

8. Why is there no comparison to online floats or dye in the LiveOcean domain of figure 6 and 7? For a complete picture, that would be useful here too.

Response: we thank the reviewer for this suggestion and tried to run online dye and particle in the LiveOcean domain. However, this requires recompiling and rerunning the model. The version of the model used for the LiveOcean domain used a somewhat dated version of ROMS (the Hood Canal model, and the current LiveOcean forecast use an up-to-date version). The result is that it would be a great deal of work, and significant computational effort, to re-run the LiveOcean model with dye.

While we agree that your suggestion would give a more complete picture, we are motivated by the difficulty of accomplishing it to explore whether it is necessary. We see no reason why intercomparisons between offline particle tracking and online particle tracking/dye experiments in the large LiveOcean domain would give significantly different information than that from intercomparisons among them in the small Hood Canal domain, i.e., that offline particle tracking performs as well as the online tracking. Our attempt here to rerun the large LiveOcean model could also be an example why offline particle tracking is more popular than online tracking in applications. Some studies that applied offline tracking can easily use the precalculated velocity fields without the necessity of rerunning the hydrodynamic model, for example, a recent study using offline particles and the global model ECCOv4 to investigate the global overturning circulation (Rousselet et al., 2021). They may not have the resources available to rerun the model.

Rousselet, L., Cessi, P., Forget, G. (2021). Coupling of the mid-depth and abyssal components of the global overturning circulation according to a state estimate. Science Advances, 7(21), eabf5478.

Other minor comments:

- line 9: Make explicit that these numbers (200m and 1000m) are the resolution and not the domain sizes)
Response: thanks for the suggestion. We deleted these two numbers in the abstract to avoid confusion. The small Hood Canal model domain has a uniform horizontal resolution of 200 m but the large LiveOcean model domain has a changing horizontal resolution from 500 m to 3000 m. In the coastal area of LiveOcean model, in which we conducted the particle tracking experiments, the grid resolution is 1000 m.

- line 15: This sentence is very vague; please rephrase in terms of conclusions/outcomes
Response: We edited this sentence lightly for clarity (lines 14-18), but were unable to boil the "tradeoffs" down to the level of conclusions. This is because the conclusion is really a user's choice based on how the tradeoffs affect their particular model and research needs.

- line 26: Explain why offline tracking is more frequently applied?
Response: we added the explanation in lines 29-31.

- Line 35: There are some recent articles that compare different tracking codes in e.g. the Agulhas: https://agupubs.onlinelibrary.wiley.com/doi/full/10.1029/2019JC015753
Response: thanks for pointing out this reference and we added it in line 49.

- line 38: what is meant here with performance? Speed? Memory? I/O? Accuracy? Reproducibility?
Response: revised in line 53.

- line 40: Is LiveOcean really 'well-established'? What does that even mean, when it comes from the developers of the model?
Response: we apologize for the unclear description and have added the development and calibrations of LiveOcean in lines 61-64.

- line 127: 'studied' instead of 'studies'
Response:  revised.

- line 145: what was the convenience why the seeding strategy was different for the online particles?

Response: in the online particle experiment, all $10^5$ particles were released inside the same model grid cell as other offline particle codes. Rather than specifying a particle distribution of 100×100×10 (longitude, latitude, vertical) inside the selected grid cell, which is easily to do in the offline codes, it requires significant coding to define a particle distribution of 100×100×10 in ROMS's online tracking file, floats.in. Therefore, we slightly adjusted the seeding strategy and specified the particles' location uniformly along the diagonal of the grid cell, which only requires 1 line of code (an example shown below). This implementation gives the same location of mass center in x, y, and z direction as other offline codes. In addition, the vertical thickness of the selected grid cell is 8.7m, about 5% of the total water depth in this location (~170m). These two release strategies should not induce significant difference to the intercomparison. We added the explanations in lines 230-232.

```
! Number of floats to release in each nested grid.  These values are
! essential because the FLOATS structure in "mod_floats" is dynamically
! allocated using these values, [1:Ngrids].
NFLOATS == 100000
! Initial floats locations for all grids:
!
!   G       Nested grid number
!   C       Initial horizontal coordinate type (0: grid units, 1: spherical)
!   T       Float trajectory type (1: Lagrangian, 2: isobaric, 3: Geopotential)
!   N       Number floats to be released at (Fx0,Fy0,Fz0)
!   Ft0     Float release time (days) after model initialization
!   Fx0     Initial float X-location (grid units or longitude)
!   Fy0     Initial float Y-location (grid units or latitude)
!   Fz0     Initial float Z-location (grid units or depth)
!   Fdt     Float cluster release time interval (days)
!   Fdx     Float cluster X-distribution parameter
!   Fdy     Float cluster Y-distribution parameter
!   Fdz     Float cluster Z-distribution parameter
POS = G, C, T, N,   Ft0,    Fx0,    Fy0,    Fz0,    Fdt,    Fdx,    Fdy,    Fdz
1  1  1  100000  0.d0  -123.0008327d0  47.5657658d0  -85.281616d0  0.d0  2.673387e-08  1.8018018e-08  8.738766e-05
```

- line 209: The point that dye spreads faster than Lagrangian particles is not new, and could be related to Markovian dynamics (I.e. dye that enters a grid cell from one side can leave it on the other side within a timestep)?

Response: thanks for pointing it out and we added references (lines 302-303) that observed the same faster dye spreading in their comparisons between dye and Lagrangian particles. For Markovian dynamics, this is an interesting topic, we assume it's related to the advection scheme of dye. Could you point us to the reference? Additionally, numerical mixing was found to account for one-third of the total mixing of salinity in the LiveOcean Model inside the Salish Sea (Broatch and MacCready, 2022), which can also contribute to the faster spreading of dye than Lagrangian particles, especially in regions with strong horizontal gradients.

Broatch, E. M., MacCready, P.: Mixing in a Salinity Variance Budget of the Salish Sea is Controlled by River Flow. J. Phys. Oceanogr., 52(10), 2305-2323. https://doi.org/10.1175/JPO-D-21-0227.1, 2022.

- line 256: 'growing differences in location' is slightly awkward phrasing?
Response: revised (line 362)

- line 278: is the laptop the same as the Apple M1 Pro?
Response: Yes, and we revised it to "Apple M1 Pro" to make it clear (line 386).

- line 284: why does LTRANS scale so poorly for large numbers of particles? This is very surprising for a Fortran code?

Response: to be honest, we don't know the exact answer. Based on our experiments, LTRANS runs very fast with small numbers of particles but slows down a lot when tracking a large number of particles. It could be due to the algorithm structures, but it is beyond the scope of this paper. Here we hope the computation time may be one piece of information scientists can use when choosing a particle tracking package and designing an experiment.

---

## Author Comment (AC4)

**GMD-2023-45 Response to Reviewers**

We appreciate the time and efforts one anonymous reviewer and Dr. Tor Nordam dedicated to providing feedback on our manuscript. We have incorporated or responded to all the comments and suggestions made by the reviewers. Please see below, in blue, for a point-by-point response to the reviewers' comments and concerns. The line numbers refer to the tracked version of our manuscript.

**Reviewer 2:**

The authors present a Lagrangian particle tracking tool, Tracker, and compare it to several other existing particle tracking codes, as well as an Eulerian transport model run as an integrated part of ROMS. In general, I find comparisons like this to be very interesting and useful. In my personal opinion, the field of Lagrangian transport modelling would probably benefit from increased attention to details of implementation and comparison between codes. Hence, the topic should (in my opinion) be of interest to the readers of GMD. However, I find that the present manuscript is a bit lacking in some aspects, and my recommendation is that it should be reconsidered after revisions.

The abstract states that Tracker is introduced, and also compared to other models. However, there are also references to previous papers that use Tracker (Brasseale & MacCready, 2021 and Stone et al., 2022), even though these papers do not seem to explicitly use the name "Tracker" (from a quick search of the documents). If the current manuscript is the definitive introduction of Tracker, it should in my opinion contain additional details on the implementation to properly document the model.

For example, are the u and v components of the current assumed to be on separate grid points, and interpolated independently? How about the vertical current component? Is variable horizontal diffusivity supported? It is stated that vertical diffusion uses reflection at the boundaries, but what about edge cases where the particle is so far outside one boundary that reflection would take it outside the other boundary (can happen with non-zero probability, since the random walk uses Gaussian numbers).

Response: we used nearest neighbor for all u,v,w interpolation. We don't use horizontal diffusivity and the assumption here is that the important motions leading to horizontal diffusion are resolved in our coastal model (the horizontal grid resolution ranges from 500 m 3000 m). If the model is not eddy resolving such as a global model, the horizontal diffusion will be needed to include the effects of eddies.

For the edge cases, we enforce limits on vertical reflection using the numpy remainder function if the vertical advection moves particles more than the total water depth. Please see line 488-508 in https://github.com/parkermac/LO/blob/v1.1/tracker/trackfun.py

I would also say that a description of the implementation would be useful for a paper introducing a new model, even if one would hope that the actual performance in terms of accuracy is independent of implementation detail. What python libraries are used? How are the particles

stored (simple arrays or custom objects?) Are particle positions recorded in lon-lat or in x-y coordinates in meters? If lon-lat, then how are displacements in meters converted to lon-lat? Are any assumptions made (and hard-coded?) about e.g. the radius of the Earth?

Response: thanks for the suggestion. We now describe more details on the implementation of Tracker in Methods (lines 113-153) and in Table 1.

Finally, I would be interested to see a bit more discussion of some of the choices that were made in the implementation. For example, why combine 4th-order Runge-Kutta with nearest-neighbour interpolation? 4th-order Runge-Kutta has requirements when it comes to continuous derivatives of the velocity field, which are not met by nearest-neighbour. Based on earlier work I have done, I would guess that you could get a better ratio of accuracy to computational effort with e.g. 2nd-order Runge-Kutta if you use nearest neigbour interpolation (see, e.g, https://doi.org/10.5194/gmd-13-5935-2020). This might not be very important in practice, though, so feel free to ignore.

Response: thanks for directing us to this study about the accuracy of different combinations of integrator and interpolation methods in calculating particle trajectories. For our case, we found that nearest neighbor gives very close results to bilinear interpolation but it scales better. It can speed up the computation in a big model grid, much faster than bilinear interpolation. We also tested the 2nd-order Runge-Kutta integration and found it performs poorly in regions with complex shoreline geometry, like the curving channels in Tacoma Narrow in the southern Salish Sea. We describe our choices of nearest neighbor and 4th-order Runge-Kutta in lines 140-144.

I would also like to see more discussion on the vertical diffusivity implementation. Why was a 3-point Hanning window chosen? And it says that this is applied in the calculation of the vertical derivative, but does that mean that the smoothing is not applied for the diffusivity values themselves? I think these values should be chosen consistently, otherwise you might have trouble with the well-mixed condition. Did you do any testing of this point? Note also that the well-mixed condition is only a neccessary condition, not a sufficient one. A perhaps more stringent test of the implementation would be to compare to a dedicated 1-D solver of the diffusion equation with variable diffusivity. The comparison to the dye in Fig. 5 is good, but since this is a 3D case with comparison only in the vertical it is a bit hard to reason about the cause of the discrepancy.

Response: we followed North et al. (2006) with the 3-point Hanning window to smooth the vertical eddy diffusivity profile and the smoothed vertical profile was then used to calculate the vertical derivative. We also tested 4-point and 8-point Hanning windows to see how well they can satisfy the vertical well mixed condition (WMC) test and found they gave similar results to the 3-point Hanning window method. In lines 132-134, we reworded the sentence to make it clear on the usage of Hanning window to smooth the vertical derivative.

The purpose of the vertical WMC tests for particles is to see if Tracker can reproduce the Eulerian solution to the 1-D vertical diffusion equation: $(\frac{\partial C}{\partial t} - \frac{\partial}{\partial z}\left(K\frac{\partial C}{\partial z}\right) = 0$, where $K$ is eddy diffusivity, with an initial uniform concentration ($C(z) = C_0$) and no flux boundaries ($K\frac{\partial C}{\partial z} = 0$). In the setup of WMC test in Tracker, we uniformly released 4,000 particles in the water column

and turned off both horizontal and vertical advection. Particles only move vertically, and the vertical location is controlled by $z_{n+1} = z_n + \frac{\partial AK_s}{\partial z} \cdot \Delta t + R\sqrt{2AK_s \cdot \Delta t}$. To satisfy the WMC test, the initially well-mixed particles are expected to remain uniform in a statistical sense regardless of the diffusivity profiles. Therefore, the WMC test applied here is like a 1-D solver of the diffusion equation. We add more details about the WMC tests in lines 157-186.

North, E. W., Hood, R. R., Chao, S. Y., Sanford, L. P.: Using a random displacement model to simulate turbulent particle motion in a baroclinic frontal zone: A new implementation scheme and model performance tests. J. Mar. Syst., 60(3-4), 365-380, https://doi.org/10.1016/j.jmarsys.2005.08.003, 2006.

The second point of the paper is a comparison of different particle models and a dye study. Here, I would have liked to see a bit more discussion and investigation of details. For example, the paper states that "a very good inter-model match was achieved", with reference to Fig. 3. However, I would say that there is something odd in Fig. 3, as the trajectories of the centers of mass already from the start appear to separate very fast. This should be straightforward to investigate further, for example by running a single timestep with a single particle and no diffusivity, and checking if all the models move the particle the same distance and direction. Certainly those models that use the same interpolation and numerical integrator should have exactly identical behaviour in the case of no diffusivity.

Response: we thank the reviewer for the interesting comment and conducted the suggested experiments by turning off diffusivity for all four offline tracking codes.

In the figures below, we used 2nd-order Runge-Kutta for OpenDrift to make its advection and interpolation schemes the same as Particulator. The time step for particle tracking is 300 s and particle locations were saved every hour. Each dot in these figures below represents every hour. We expected OpenDrift and Particulator should give the same trajectory after turning off diffusion, but they sort of separate from each other in the first hour after 12 steps of tracking. Even though we selected the same interpolation and integration methods in different particle tracking codes, we are not sure if their interpolation/advection are implemented in exactly the same way. It is beyond the scope of the paper to evaluate all the numerical details. The roundoff error for different programming language might also contribute to the difference. Whatever the source of small differences, the differences generally accumulate over time as particles experience different advection and mixing, making the comparison ever less "direct" as time goes on. This decorrelation scale is an intrinsic problem of Lagrangian analysis.

Additionally, if two particle tracking codes applied exactly the same numerical methods and the same algorithm structures, we will expect them to give the same results, but if so, the results are not of any research interest. In this study, we are trying to look at several commonly used tracking codes that have many differences, but all of them perform similarly when tested in the same circulation model. The comparison of multiple tracking codes in the same circulation model output establishes confidence in all, and allows comparison of other factors such as their computational speed and ease of use.

[Figure]

Laminar(solid) v.s. turbulence(dashed), one particle: #5000

Laminar(solid) v.s. turbulence(dashed), one particle: #50490

It is further stated that "vertical distributions of particles (Figure 5) exhibit similar evolutions among all tracking codes", whereas I would say that the distributions are quite different. In particular, Tracker shows quite a spikey distribution, and also the dye study has a lot of spikes in the distribution. This last point was particularly surprising to me, as I would have expected an Eulerian diffusion solver to produce a smoother concentration field than a particle based model. Of course, as this is 3D with advection, that makes it a bit harder to reason about, but this could be investigated. In any case, I would say that it is clear from the left panel of Fig. 5 that these models are _not_ equivalent, or at least not run with equivalent setups.

Response: we agree with the reviewer that there models are not run with equivalent setups because they used different interpolation and integration schemes. They also have different ways to process vertical diffusivity.

For the spikes in the vertical distribution of dye, the 3D advection-diffusion process could be one reason. Another reason could be due to the way we did the post-processing of the vertical position of dye (now described in greater details in lines 296-300). To obtain the histogram of vertical dye distribution, dye mass inside each model grid cell was first converted to an equivalent number of particles. Then the vertical coordinate in the center of the grid cell that

contains dye was used to represent the vertical location of dye-converted particle number. This conversion might have led to the spiky vertical distribution in the early stage of dye transport as seen in Figure 5a.

Looking at Figs. 6 and 8, I am a bit surprised about the large vertical fluctuations in position, particularly in the mid-watercolumn release. Panel g in Fig 6 shows that with Tracker, the center of mass moves more than 60 meters downwards in less than 12 hours, which is a much larger displacement than with any of the other models. I'm also curious about what the mechanism behind this transport is. How large is the vertical current component? How stratified is the water column? Discussion on this point would be appreciated.

Response: we thank the reviewer for pointing this out. In the figure below, we plotted the vertical center of mass for particle trajectories calculated from each offline codes, the vertical velocity of the model grid cell that includes the center of particle mass, and the bottom and surface salinity difference (dS) of the water column. The large downward transport in Fig. 6g happened around hours 13-22 and the averaged vertical velocity at the center of particle mass is -1.8×e-3m/s (Tracker), -1.6e-3m/s (OpenDrfit), -3.6e-4m/s (Particulator), and -2.3e-4m/s (LTRANS). Negative values mean downward. The stratification is also weaker for particles tracked by Tracker, which may because the horizontal center of mass calculated from Tracker is closer to the coastline during this period (Fig. 6h). Therefore, large downward vertical velocity and weaker stratification could lead to the large vertical displacement of particles tracked by Tracker. Generally, if the horizontal advection (due to different interpolation and integration methods) leads particles to different places, they might experience more (or less) vertical advection (lines 348-353).

[Figure]

Finally, I think it would be good in the interest of reproducibility to provide the actual setups used for the different models, perhaps in a separate github repo for the paper. That would make it much easier for others who might want to look into the comparison.

Response: we thank the reviewer for the suggestion, yet it is very challenging to upload the test data (~200 GB in total) to an online repository. For reproducibility, it may be more relevant for the interested readers to set up their own particle tracking experiments using all these open-access tracking codes. In this study, as stated in lines 199-202, we only consider the advection of passive particles and vertical turbulence (without any particle behaviors), which are generally the most basic set up for a particle tracking model.

---

## Editor Decision (ED1)

**LPT_intercomparis...**

| Name | Date Modified | Size | Kind |
|------|---------------|------|------|
| .git | 29 November 2023 at 20:00 | -- | Folder |
| LICENSE | 29 November 2023 at 20:00 | 1 KB | Document |
| LTRANS | 29 November 2023 at 20:00 | -- | Folder |
| OpenDrift | Today at 12:23 | -- | Folder |
| Particulator | 29 November 2023 at 20:00 | -- | Folder |
| README.md | 29 November 2023 at 20:00 | 904 bytes | Markdown Text |
| ROMS_online_floats | Today at 11:24 | -- | Folder |

---

## Author Response (AR3)

**GMD-2023-45 Response to Reviewers**

We appreciate the time and efforts the two reviewers dedicated in the second round of review to providing feedback on our manuscript "*Intercomparisons of five ocean particle tracking software packages in the Regional Ocean Modeling System*". We have incorporated or responded to all the comments and suggestions made by the reviewers. Please see below, in blue, for a point-by-point response to the reviewers' comments and concerns. The line numbers refer to the tracked version of our manuscript.

**Reviewer 1:**

Review of the revised version of "Intercomparisons of five ocean particle tracking software packages" by Xiong and MacCready.

I thank the authors for their extensive and detailed replies to my queries and comments. While many of the issues that I raised have been solved in the revised version of the manuscript, I am left with the most important one (original comment 1): what does this new Lagrangian code add? Or, in other words, what niche does it fill? The authors do not sufficiently answer that, in my opinion. In the reply, they discuss comparison to other codes, but in a diverse ecosystem of Lagrangian codes, each one has its niche (at least when it was first developed). I don't see that niche for Tracker: what it can do that other codes can't.

I leave it to the editor whether this manuscript, despite my reservation, still meets the bar for Geoscientific Model Development; or whether it would be better suited for a (Diamond Open Access) journal like the Journal of Open Source Software.

**Response:** we thank both reviewers' comments on the niche of Tracker that inspire us to clarify the uniqueness of Tracker and emphasize the major findings of our study. We think that the most unique thing about Tracker, compared to other packages compatible with ROMS modeling system, lies in its much faster execution time, a feature attributed to the efficiency of the nearest neighbor searching algorithm. This performance enhancement is especially pronounced in the large model domain we tested (please see Figure 9).

Within a forecast system such as LiveOcean, a reasonable computational burden is important when the offline particle tracking based on forecasted hydrodynamics can be finished in a timely manner. Even though Particulator (one of the tracking packages that were tested in this study and is written in MATLAB) can run fast with millions of particles, the commercial software MATLAB is less accessible than the open-source Python. Therefore, besides saving computation time, platform independence is another unique thing about Tracker (Table 1, under the category of 'Ease of use').

In addition, the most important finding in this study is that although all tested particle tracking packages have different choices in e.g., interpolation schemes, advection schemes, boundary conditions, or compatibilities with different numerical models or forcing data sources that make them unique, they do not end up with very different results.

In the revised (tracked) manuscript, we added these statements in the sections of abstract (lines 13-14, 18), section 3.4 (lines 313-325), and conclusions (lines 351-352).

**Reviewer 2:**

In general, I find the paper to be improved and with many added details. I think the paper should be of interest to the community, and I recommend publication after some minor revisions.

Regarding the use of the 3-point Hanning window: As far as I can tell, North et al. do not in fact use this approach, but rather discuss a 4-point and 8-point moving average. If this is the case, then I would suggest adding a reasonably clear explanation of what a Hanning window is, and how it is used to smooth a function. Or alternatively, a reference to a clear explanation. I think this is an important point to make clear, as this is one of the few implementation details where the different models compared are known to be different.

**Response:** we thank the review for pointing out lack of details for the 3-point Hanning window method. More explanations and the reference to this method were added in lines 98-101 in the tracked manuscript. The implementation of 3-point Hanning window can also be found in line 188 in trackfun.py (https://github.com/parkermac/LO/blob/main/tracker/trackfun.py).

At lines 99-100 in the revised manuscript, you state that Eqs. (1) - (3) are solved with 4th-order Runge-Kutta. However, these equations are not written as ordinary differential equations, but rather already presented as discrete numerical schemes. Specifically, Eqs. (1) and (2) are the forward-Euler representation of the horizontal transport equation, and Eq. (3) is the Visser-scheme implementation of the stochastic differential equation for vertical transport (which can in any case not be solved by 4th-order Runge-Kutta, as that is an ODE method, not an SDE method). This should be corrected, and it would also be useful to clarify here if the same timestep is used for horizontal and vertical motion, as it is fairly common to use a shorter timestep for the vertical transport. See for example discussions on the difference between horizontal and vertical equations and timestep in North et al. (2006) or Rowe et al. (2016), both of which are in your list of references.

**Response:** thank you for the suggestions and we apologized for the unclear statement. The velocity components (u, v, w) used in Equations 1-3 are obtained using the 4$^{th}$-order Runge-Kutta scheme. We corrected the statement in lines 103-105 in the tracked manuscript.

We also clarified that the same timestep is used for horizontal and vertical motion (lines 105-108). The random displacement model criterion requires dt << min($1/abs(AK_s'')$) (Visser, 1997; North et al. 2006; Rowe et al. 2016). $AK_s''$ is the second derivative of vertical diffusivity. We examined the profiles of $1/abs(AK_s'')$ (please see figures below, the x-axis is log10 scaled) based on the hourly $AK_s$ output at the six stations that were selected to test the vertical well-mixed conditions. $AK_s$ was smoothed using a 3-point Hanning window. The red dash line denotes the timestep of 300s used in Tracker and other offline tracking packages. In most occasions, dt << min($1/abs(AK_s'')$) could be achieved in the model.

[Figure]

[Figure]

Lines 127-130: The approach to finding the "significant range" for the WMC test is perfectly fine, but just as an observation (feel free to ignore), it is possible to make this a bit more "rigorous". If you consider any one of the 28 bins, then the probability of a particle ending up in that bin (when positions are drawn uniformly) is 1/28. Hence, drawing 4000 particles and counting the number in a bin is equivalent to drawing a number from a binomial distribution with 4000 trials and probability of success 1/28. The mean of such a distribution is 4000 * (1/28), and the variance is 4000 * (1/28) * (27/28). If you let the "significant range" be for example the mean plus or minus two times the square root of the variance, then you expect the number of particles in that bin to be outside the range about 4.5% of the time.

**Response:** we appreciate the reviewer's idea and checked that by this way, the mean += 2*sqrt(var) = 119 ~ 166, which is close to 102.5~187.3 used in Figure 2. Therefore, we would like to keep our approach to finding the significant range for the WMC test.

Line 150: It says "decrease their variance", but it should perhaps be "increase"?

**Response:** it is actually "decrease". We reworded this sentence to make it clear (lines 158-159)

Lines 226-230: It is not clear why you chose to convert the concentration in ROMS to "an equivalent number of particles", at the center of the cell. Is it not better, easier and more accurate to present the horizontally integrated concentrations directly?

**Response:** We agree with the review that it will be more accurate to present the horizontally integrated contractions of dye directly and will give a smoother vertical distribution in Figure 5. Here, in converting dye mass to an equivalent number of particles, our intention was to give dye and particle distributions from different particle tracking packages the same unit, i.e., particle number, so that they could be compared directly in Figure 5.

Finally, regarding the data: I believe GMD has a data availability policy that includes such things as "data sets for forcing of models". (https://www.geoscientific-modeldevelopment.net/policies/code_and_data_policy.html). You say in the reply to reviewers that the dataset is around 200 GB, but it should be possible to reduce the size quite significantly by "cropping" (with ncks, for example) the data horizontally to include only what is needed to reproduce the results. The model domain shown to the left in Fig 1 appears to be 1000 or so from south to north, but the longest example trajectories shown only need a rectangle of data of about 160 km by 120 km, which should reduce the size a fair bit.

**Response:** we appreciate the reviewer's constructive comments and uploaded our code and exemplary hydrodynamic outputs to https://github.com/Jilian0717/LPT_intercomparison/tree/main and https://zenodo.org/records/10223144. These two links were also added in the section of Code/Data availability.

Instead of trying to make model extractions to fit the particle tracks, we took the approach of including a subset of the ROMS history files that could be used with our code. We hope that this is sufficient.

Additional private note (visible to authors and reviewers only):
In what conditions/circumstances should Tracker be used instead of any other model and why?

**Response:** please also see our response to reviewer 1 above. Generally, Tracker runs faster than other particle tracking packages that were tested in the present study, especially in the large model domain (please see Figure 9). In this study, we demonstrated that all tracking packages, although with different specialties, would give very similar results. Therefore, if computation time is the foremost factor, for example, in a forecast system with a large domain size like LiveOcean and millions of particles are required, Tracker would be a good choice. In Figure 9, we also observed that Particulator (written in MATLAB) runs very fast with one million particles, yet MATALB is a commercial software that requires not cheap license, thus much less accessible than Python.

---

## Author Response (AR4)

GMD-2023-45 Intercomparisons of five ocean particle tracking software packages in the Regional Ocean Modeling System

We appreciate suggestions from the topic editor to make our code and data more accessible to the public. Please see below, in blue, for a point-by-point response. The line numbers refer to the tracked version of our manuscript.

**Topical editor's comments:**

**Public justification (visible to the public if the article is accepted and published)**:
1) There are several instances of references to other models e.g.: "OceanParcels (https://oceanparcels.org/), Ichthyop (https://ichthyop.org/), TRACMASS (Döös et al., 2013), PaTATO (Fredj et al., 2016), TrackMPD (Jalon-Rojas et al., 2019), OceanTracker (Vennell et al., 2021), Deft3D-PART (Deltares, 2022), Ariane (http://stockage.univ-brest.fr/~grima/Ariane/), and CMS (https://github.com/beatrixparis/connectivity-modeling-system)."

Please provide DOIs when possible or last access dates to the webpages.

**Response:** we replaced all links to the webpages with respective references that have DOIs (lines 33-36).

2) When referring to unpublished material, e.g.:
"https://faculty.washington.edu/pmacc/LO/p5_Phab_full_salt_top.html,..." one would usually provide last access date. It would be even better to provide a screen capture of the page as pdf in the supplements. In the long run this page will be probably lost.
This applies to all web citations.

**Response:** thanks for the reminder. We replaced this link with an appropriate citation here (lines 47-48). The link in line 57 has been replaced with a screen capture of the page as pdf in the supplements, and the link in lines 154-155 was replaced with a reference.

3) Throughout the text there are a references to your own work hosted in github. The GMD policy does not accept github as a permanent resource. You need to provide a referenced to Zenodo (with DOI) or submit the git directory as supplementary material.
(an example can be found in https://github.com/parkermac/LO/tree/v1.1/tracker/experiments.py).
Please develop the example and provide all necessary files to run it as supplement.

**Response:** we removed the links to Github (lines 120-123) and attached the two codes (experiments.py and zfun.py) in the supplementary material.

4) More importantly, the Code availability section (line 360) expressess: "All example codes and

hydrodynamic outputs that were used to test these five particle tracking packages can be found in https://github.com/Jilian0717/LPT_intercomparison/tree/main and https://zenodo.org/records/10223144."

Checking the Zenodo zip archives, one can find the "model_output" folder, that contains the advecting fields and initial conditions for the models and the "LPT_intercomparison_tracking_packages" that contains the code for some of the models. It doesn't contain a subfolder for the source code of Tracker though.

Please make sure that all necessary code, input data, documentation and instructions (tutorials) are included in the Zenodo repository or, even better, as supplementary material to this submission.
The archive should be self contained without references to external websites.
It should be possible to install and run the models (at the very least Tracker) and produce the relevant output following specific instructions.

**Response:** Thanks for the suggestion. It was not possible to upload all code and input data as supplementary materials due to file size limitations, however we did include representative samples of the model output files to allow the archived codes to run. Please see the new Zendo repository at https://doi.org/10.5281/zenodo.10493839. We revised the statements in Code/Data availability (lines 359-366) as well.

---

## Author Response (AR5)

GMD-2023-45 Intercomparisons of Tracker v1.1 and four other ocean particle tracking software packages in the Regional Ocean Modeling System

We appreciate the time and efforts the topic editor made to improve our manuscript. Please see below, in blue, for a point-by-point response. The line numbers refer to the tracked version of our manuscript.

**Topical editor's comments:**

**Public justification (visible to the public if the article is accepted and published)**:
Since the manuscript is in the category "Model description papers", please make sure the appropriate rules are follwoed (https://www.geoscientific-model-development.net/about/manuscript_types.html#item1)
In particular:
- The main paper must give the model name and version number (or other unique identifier) in the title.
- The publication should consist of three parts: the main paper, a user manual, and the source code, ideally supported by some summary outputs from test case simulations.

So please include the name and version of Track in the title and abstract.

**Response:** Thanks for the suggestion. We included the name and version of Tracker in the title and abstract. The title was changed to "Intercomparisons of Tracker v1.1 and four other ocean particle tracking software packages in the Regional Ocean Modeling System". Please see revisions in lines 1 and 7 in the clear manuscript.

I have revised the updated Zenodo package and I didn't get to run the first example.

LPT_intercomparison_tracking_packages/Tracker$python3 make_KDTrees.py -gtx cas6_v3_lo8b -d 2019.07.04 -ro 2
Traceback (most recent call last):
File "/LPT_intercomparison_tracking_packages/Tracker/make_KDTrees.py", line 19, in
from lo_tools import Lfun, zrfun
ModuleNotFoundError: No module named 'lo_tools'

I cloned LO
git clone https://github.com/parkermac/LO.git
and made the required module available to python and run in a subsequent error:
Error from Lfun: missing LO/get_lo_info.py and LO_user/get_lo_info.py

Please make sure the examples run straightforwardly for a new user.

Ideally you should provide working examples of all the models used, but it would be best that at

least the Tracker examples work out of the box. Otherwise provide working instructions for the dependencies.

**Response:** Thank you very much for checking the codes. In the revised Zenodo repository, we generated a new simpler standalone example for Tracker v1.1 so that the readers can run it without the necessity to install the customized python environments or link to any directories specific to our own LiveOcean system. The updated Zenodo repository can be found in https://doi.org/10.5281/zenodo.10810102 and in line 354 of the clear manuscript.

In the README.md file: what do you mean with "This code uses nearest neighbor for most everything and so might work with more complex ROMS grids (the LO grids are plaid), but that is untested."?

**Response:** We removed the README.md file, which contains this statement, in the folder containing codes and hydrodynamic files of Tracker. Instead, we generated a concise README.md file in the new archived Zenodo/LPT_intercomparison_tracking_packages/Tracker_v1.1/.

To answer the editor's question: in the original README.md, we intended to inform other users who may have, for example, a curved ROMS model grid that the nearest neighbor algorithm used in Tracker should work for their grids. Since this is untested, we removed the sentence from the revised README.md.

Besides the above response, we also made some minor Language edits in the revised manuscript.